# Meta-analysis reveals that phenotypic plasticity and divergent selection promote reproductive isolation during incipient speciation

Benjamin J. M. Jarrett ●[1,2] ✉, Philip A. Downing ●[2,3] & Erik I. Svensson ●[2]

The evolution of reproductive isolation is a key evolutionary process, but the factors that shape its development in the early stages of speciation require clarification. Here, using a meta-analysis of 34 experimental speciation studies on arthropods, yeast and vertebrates, we show that populations subject to divergent selection evolved stronger reproductive isolation compared with populations that evolved in similar environments, consistent with ecological speciation theory. However, and contrary to predictions, reproductive isolation did not increase with the number of generations. Phenotypic plasticity could partly explain these results as divergent environments induce a plastic increase in reproductive isolation greater than the effect of divergent selection, but only for pre-mating isolating barriers. Our results highlight that adaptive evolution in response to different environments in conjunction with plasticity can initiate a rapid increase in reproductive isolation in the early stage of speciation.

The evolution of reproductive isolation is a crucial process both in initiating and completing speciation[1]. Rooted in the biological species concept[2,3], reproductive isolation can be viewed as an emergent property of an interaction between two populations that results in a restriction of gene flow between the same populations[2]. Traditionally, such reproductive isolation is thought to be exclusively caused by inherited differences between populations[2], though how to measure reproductive isolation and its role in completing speciation is still debated[4–7]. A long-standing explanation for the evolution of reproductive isolation is the ecological speciation model[8–10], whereby reproductive isolation evolves as a by-product of natural selection promoting different phenotypes in different environments[8–10]. Adaptation in response to divergent selection imposed by different environments can result in greater divergence in inherited factors, such as alleles or symbionts, between populations than populations experiencing similar selective regimes. This increased divergence between populations increases the likelihood that other inherited factors accelerate the rate of evolution of reproductive isolation between the populations. For example, barrier loci may hitchhike with alleles at other loci that are subject to divergent natural selection in the different environments[8,9]. Alternatively, selected loci could pleiotropically promote reproductive isolation[8,9]. With its roots in Darwin's work[11], research during the past three decades has documented several examples in some well-studied taxa, which strongly implicate a key role of divergent selection in speciation[9,12–14].

But how general of a mechanism is ecological speciation? An alternative mechanism, also driven by selection, is mutation-order speciation, whereby populations experience similar selection pressures, but adaptations may be underpinned by the same, or by different, mutations in the different populations. The order in which mutations fix in each population can then contribute to the evolution of reproductive isolation as these mutations may be incompatible with each other, causing hybrid inviability or sterility via Dobzhansky–Muller incompatibilities[1,13]. Species adapting to similar environments are predicted to show similar phenotypes in ecologically relevant traits,

[1]School of Environmental and Natural Sciences, Bangor University, Bangor, UK. [2]Department of Biology, Lund University, Lund, Sweden. [3]Ecology and Genetics Research Unit, University of Oulu, Oulu, Finland. ✉e-mail: b.jarrett@bangor.ac.uk

such as body size and mouthparts; a prediction used by Anderson and Weir[15] in a recent comparative analysis designed to quantify the relative contribution of ecological speciation and mutation-order speciation mechanisms in a number of vertebrate species. They found that sister species were about ten times more likely to be phenotypically similar to each other, which the authors interpret as evidence for sister species evolving in similar environments[15]. This intriguing result suggests that mutation-order speciation could be the dominant speciation mechanism in vertebrates and ecological speciation might only explain a minority of speciation events[15].

While comparative analyses are powerful to illuminate the processes that underpin the generation of new species[5,15–22], they are not our only tool for studying speciation. Experimental evolution of reproductive isolation is a complementary tool, starting from the opposite end of the speciation process—at its beginning[23,24]. Experimental evolution enables researchers to causally test mechanisms behind the evolution of reproductive isolation by dividing a single population into different replicates, imposing different experimental regimes to these replicates and then estimating reproductive isolation after any number of generations (Fig. 1a,b). While experimental evolution probably cannot replicate the entire speciation process, from initial divergence to the formation of true species, rapid adaptation and speciation often relies on standing genetic variation[25,26], meaning that the early phase of speciation can be mimicked in the laboratory[10]. Laboratory speciation experiments started in the 1950s, with researchers testing mechanisms of speciation such as reinforcement, divergence with gene flow, bottlenecks and the allopatric speciation models. These early studies were synthesized by Rice and Hostert[10] in a now classic review. One conclusion from their synthesis was that divergent selection promoted the evolution of reproductive isolation via pleiotropy and/or genetic hitchhiking[10]. This conclusion stimulated much later research into the role of divergent selection to promote reproductive isolation and provided a conceptual grounding for the theory of ecological speciation[8,9]. Lacking from Rice and Hostert[10], however, was a quantitative analysis of the strength reproductive isolation between populations evolving in divergent environments relative to that between populations evolving in similar environments. Here, we investigated this using a formal meta-analysis to quantitatively assess the effect of divergent selection in promoting reproductive isolation and examine the factors that influence its evolution in the early stage of speciation.

## Database of experimental speciation experiments

We performed a literature search for experimental speciation experiments published in the 30 years after the publication of Rice and Hostert[10] (Methods). Rice and Hostert[10] stimulated several new experimental evolution studies testing causal mechanisms underlying the evolution of reproductive isolation[23,24], now enabling our formal meta-analysis. Though multiple mechanisms have been investigated in many experimental evolution studies, including the roles of genetic bottlenecks and sexual conflict, here we focus on if and how divergent selection could increase the rate of evolution of reproductive isolation. We compiled a dataset of 1,723 effect sizes from 34 studies on 15 species that imposed divergent selection and estimated reproductive isolation within and between selective regimes (Methods). Estimates of reproductive isolation within environments allow us to quantify the role of mutation-order speciation; that is, populations that evolve reproductive isolation in response to the same environment (Fig. 1a). Estimates of reproductive isolation between populations subject to different selection regimes allow us to quantify the role of ecological speciation (Fig. 1b).

As effect sizes, we used the metric of Sobel and Chen[27], which quantifies reproductive isolation between populations as

$$RI = 1 - 2 \times \left( \frac{H}{H+C} \right),$$

where $H$ is the number or frequency of heterospecific/heterotypic matings and $C$ is the number or frequency of conspecific/homotypic matings. This framework can be readily applied for both pre-mating and post-mating isolation where $H$ can be substituted for immigrant or hybrid fitness and $C$ for resident fitness. This equation can be expanded to include any isolating mechanism separately and place it on the same scale of −1 to +1, where +1 is complete reproductive isolation, −1 is complete gene flow between the two populations (complete disassortative mating, for example) and 0 is random gene flow between populations. As this metric is equivalent to Pearson's correlation coefficient, we used Fisher's $z$-transformation to normalize it ($z$RI). We used Bayesian models to estimate reproductive isolation from each paper, with the square of the standard deviation of the posterior distribution of $z$RI as the sampling variance (see Methods for sensitivity analyses using a different estimate for the sampling variance).

## Results and discussion

Unsurprisingly, invertebrates dominated the dataset (11/15 species; Fig. 1c,d and Supplementary Table 3), with *Drosophila* species making up the majority (32.4%) of the experiments, as in Rice and Hostert[10]. Here, we fit a full meta-analytic model including all the variables subsequently listed. We classified each estimate of reproductive isolation as a pre-mating or post-mating barrier. Pre-mating isolation barriers were reported more commonly than post-mating barriers, representing 91.7% of the total effect sizes, and sexual isolation being the most common pre-mating barrier estimated (see Methods for a detailed breakdown of the effect sizes). We also collected data on (1) how many generations the experiment had been running when reproductive isolation was estimated (median of 43 generations and range of 8–1,589), (2) founding population size for each replicate (median of 280 and range of 1–5,000), (3) whether populations experienced a common garden generation to minimize environmental effects on the estimate[28], (4) whether estimates were at the replicate population level or if estimates involved more than two populations (Methods) and (5) if our estimate of reproductive isolation was calculated from simulated data estimated from means and standard errors from papers where raw data were not available.

Finally, we collected information on whether reproductive isolation was measured between populations evolving in the same environment, or whether the two populations experienced divergent selection imposed by different environments. If divergent selection and ecological differentiation increase the rate of evolution of reproductive isolation, reproductive isolation would be expected to be greater between populations evolving in different environments compared with populations evolving in the same environment.

### Divergent selection increases reproductive isolation

Populations that evolved in different environments exhibited greater reproductive isolation than populations that had evolved in the same environment (difference estimate = 0.073, 95% credible intervals (CIs) = [0.037, 0.103], pMCMC = <0.001; Fig. 2a). This result aligns with the predictions made by ecological speciation theory[9] and with the initial conclusions reached by Rice and Hostert[10] three decades ago. Adaptation to different environments probably accelerates the build-up of inherited differences between populations, which increases the likelihood that alleles that contribute to reproductive isolation are linked with selected alleles[29]. Alternatively, selected alleles pleiotropically contribute to reproductive isolation[30]. We note that divergent selection accelerated the evolution of reproductive isolation even with a reduced dataset (950 effect sizes from 24 studies) that only included effects sizes estimated after a common garden generation (difference estimate = 0.073, 95% CI = [0.040, 0.107], pMCMC < 0.001). These results suggest that the evolution of reproductive isolation by divergent selection may be common and detectable across a range of taxa and environments.

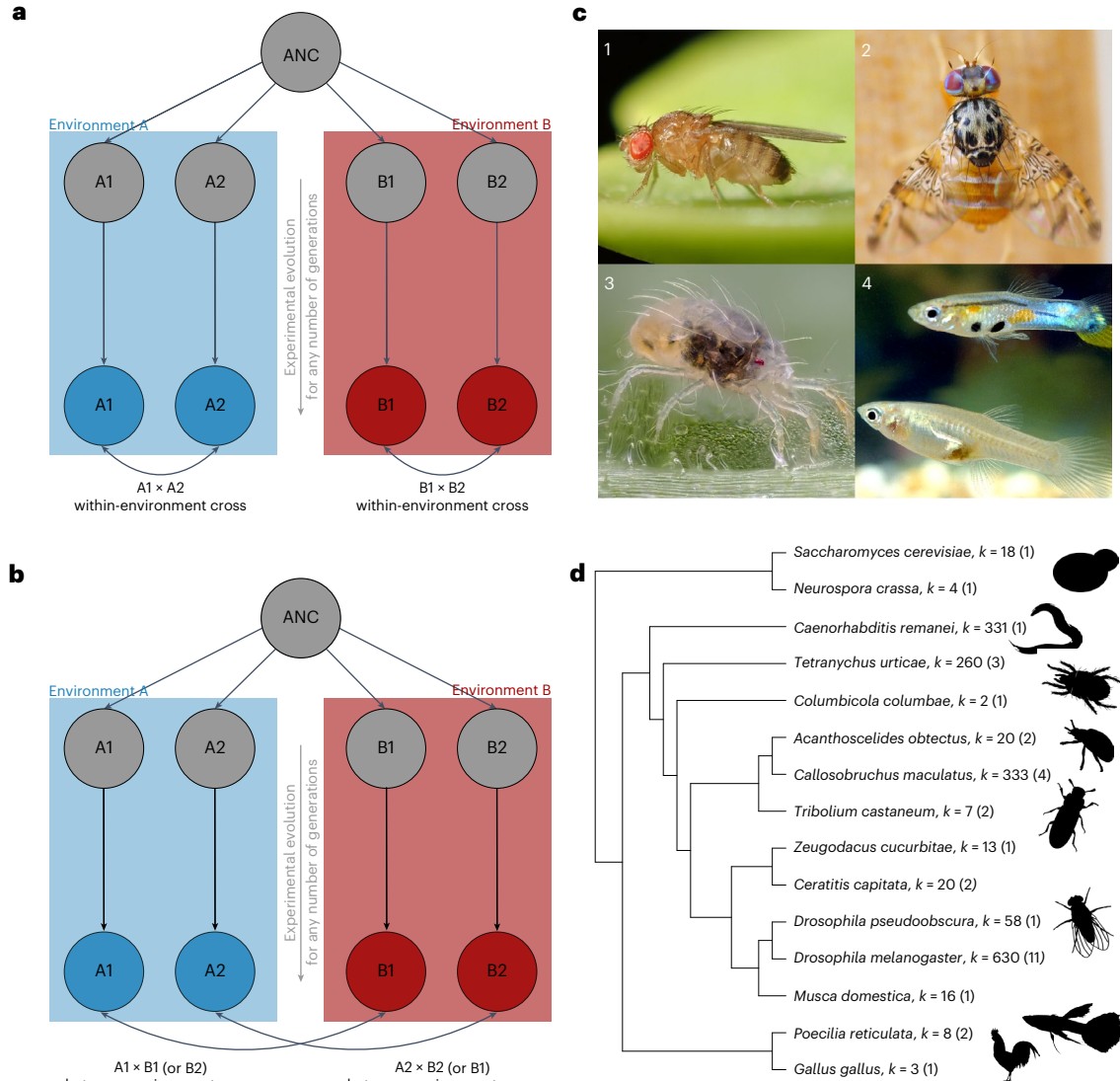

**Fig. 1 | An overview of the type of experiment and effect sizes, and species included in the dataset. a**, An outline of the experimental design we sought: a single ancestral (ANC) population is split into two or more replicate populations that are exposed to different environments or selective regimes (blue or red). Within-environment crosses are obtained between replicates evolving in the same environment (for example, A1 × A2 and B1 × B2). **b**, Between-environment crosses are obtained between populations evolving in different environments (for example, A1 × B1, A1 × B2, A2 × B1 and A2 × B2). If estimates of reproductive isolation are greater between environments than within environments, this is evidence for ecological speciation, whereas if they are equal, this provides evidence for mutation-order speciation. **c**, The dataset is predominantly comprised of data from invertebrates that are best suited for experimental evolution studies, like *Drosophila melanogaster* (image 1),

*Ceratitis capitata* (image 2) and *Tetranychus urticae* (image 3), though some vertebrate translocation experiments, such as those done in *Poecilia reticulata* (image 4) also fit the criteria. **d**, The phylogeny of all the species included in the dataset, from yeast to nematodes to vertebrates, with the number of effect sizes (*k*) (and studies in parentheses) each species contributes towards the meta-analysis. Photos in **c** reproduced with permission from: 1, Alexis Tinker-Tsavalas; 2, Daniel Feliciano; 3, Gilles San Martin; 4, ref. 122, under a Creative Commons license CC BY 4.0. Silhouettes in **d** reproduced from PhyloPic under a Creative Commons license CC0 1.0: *Sophophora melanogaster*, Andy Wilson; *Gallus gallus*, Steven Trver; *Poecilia reticulata*, seung9park; *Tribolium castaneum* and *Tetranychus urticae*, Christoph Schomburg; *Acanthoscelides obtectus*, Birgit Lang; *Caenorhabditis elegans*, Jake Warner; *Saccharomyces cerevisiae*, Wayne Decatur.

Diet was the predominant type of environment and form of divergent selection in these experimental evolution studies (55.9% of studies; Supplementary Information). While diet may select on a range of phenotypic traits, most studies actively manipulated a single dimension of divergent selection. Two exceptions were Castillo et al.[31] who manipulated both diet and access to food, and Rundle[32] who used ancestral and novel environments that differed in diet, temperature, photoperiod and feeding schedule. While our data suggest that a single environmental dimension of divergent selection is sufficient to promote the evolution of reproductive isolation, multidimensional or multifarious selection has been suggested to accelerate adaptation[10,33]. However,

there is currently little direct empirical data to support the claims that multifarious selection can accelerate the evolution of reproductive isolation[34,35]. Manipulating multiple environmental axes to generate multifarious divergent selection should be a high priority in future experimental speciation studies as changes in ecology are likely to be multidimensional[24].

## Reproductive isolation does not increase with time

A central tenet of speciation research is that reproductive isolation should increase with time as two populations or incipient species diverge[1]. The shape of the relationship between reproductive isolation

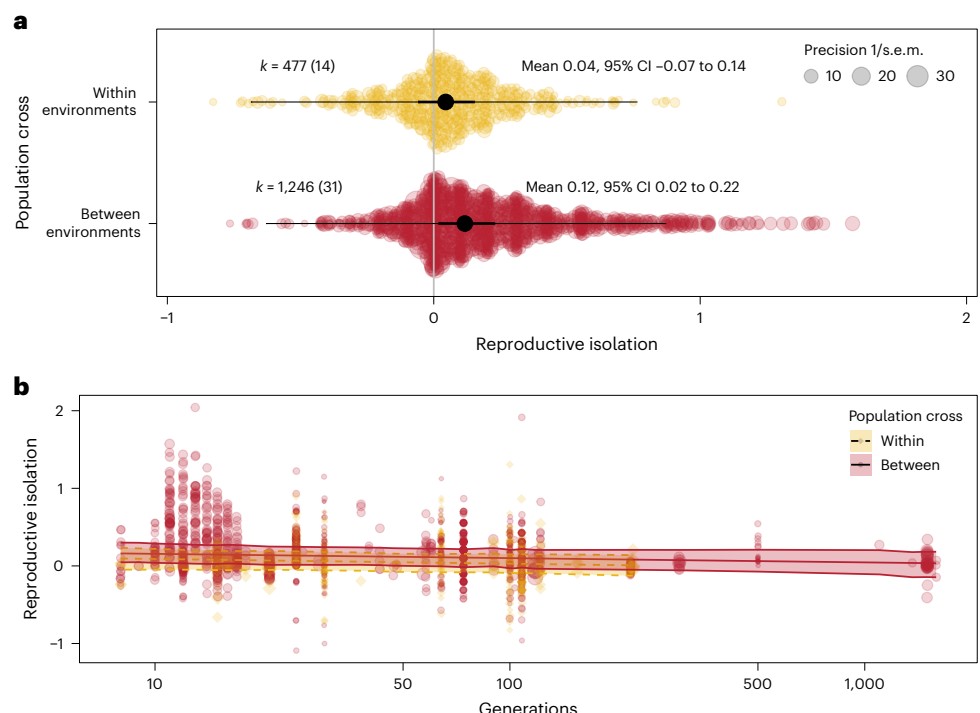

**Fig. 2 | Meta-analytical and experimental evidence for the evolution of reproductive isolation as a by-product of divergent selection. a**, Reproductive isolation is greater for populations that have been evolving in different environments (between environments, red) compared with populations that have been evolving in the same environment (within environments, yellow). *k* indicates the number of effects sizes extracted with the number of studies in parentheses. The thick black lines are the 95% CIs and the thin black lines are the prediction intervals. **b**, However, the magnitude of reproductive isolation is independent of the number of generations of the experimental evolution study. Reproductive isolation is *z*-transformed. Each point is an effect size (yellow diamonds for within-population crosses and red circles for between-population crosses). The size of the point indicates the precision (1/s.e.m.) of the effect size, as shown in the legend in **a**. The lines represent the posterior predicted mean of reproductive isolation, with the shaded area indicating the 95% CI (dotted yellow lines are within-population crosses, and solid red lines are between-population crosses).

and time (or genetic distance) would depend on the mechanism underlying reproductive isolation. To understand the dynamics of reproductive isolation at the onset of speciation, we tested for the presence of an interaction between the population cross (within or between environments) and the number of generations of evolution. Our prediction was that the between-environment slope of how reproductive isolation develops with time would be greater than the within-environment slope, leading to the end result that divergent selection increases reproductive isolation (Fig. 2a). Surprisingly, we did not find evidence of an interaction (slope difference estimate = $-1.40 \times 10^{-7}$, 95% CIs = $[-6.37 \times 10^{-4}, 5.70 \times 10^{-4}]$, pMCMC = 0.98), nor did we find an effect of the number of generations at all (slope estimate = $-1.40 \times 10^{-5}$, 95% CIs = $[-1.45 \times 10^{-4}, 1.29 \times 10^{-4}]$, pMCMC = 0.89). Reproductive isolation, therefore, does not increase with time in these experimental speciation studies (Fig. 2b).

One possible explanation for this unexpected result is that the environmental manipulations designed by experimenters induced strong selection in the early stages of the experiment so that adaptation and concomitant reproductive isolation developed rapidly and almost instantaneously. The major part of reproductive isolation would then be expected to have evolved in the earliest stages of the experiments, when populations were maladapted to their novel environments. This hypothesis is difficult to test as first, estimates of the strength of selection were not available for these studies and second, reproductive isolation was usually measured once or twice during the course of a single experiment. Therefore, the temporal dynamics of reproductive isolation may be obscured by the different experimental study designs. To circumvent this problem, we analysed a subset of our data containing those studies that measured reproductive isolation at three or more timepoints (*N* = 5 studies; Supplementary Information). The average

slope was not significantly different from zero in this this subset of studies either (slope estimate = $-0.224$, 95% CIs = $[-1.046, 0.526]$, pMCMC = 0.54), but the slope estimate tended to be negative, which is consistent with reproductive isolation emerging early in the experiments when selection is strong and populations are still relatively maladapted.

A second possible explanation is that the strength of selection was not of equal magnitude across studies. This would have resulted in different rates of adaptation and thus unequal probabilities of the evolution of reproductive isolation. Such between-study heterogeneity may have obscured a clear pattern between reproductive isolation and the number generations. A third possibility could be that populations with smaller effective population sizes may have fixed alleles faster due to genetic drift, compared with larger populations. This would lead to forestalled adaptation and a cessation of the gradual build-up of reproductive isolation. To test this hypothesis, we examined the founding population size reported by each study. However, we found no significant relationship between founding population size and the rate of evolution of reproductive isolation (slope estimate = 0.004, 95% CIs = $[-0.013, 0.019]$, pMCMC = 0.708).

## Plasticity increases pre-mating reproductive isolation

A fourth explanation is that phenotypic plasticity induced by the divergent environments causes reproductive isolation itself within a single generation[36-38]. This may be particularly true for pre-mating isolating barriers where the early developmental environment, such as diet[39] or host plant[40,41], can alter sexual signals and the preferences for these same signals. Post-mating barriers, however, may be less influenced by plasticity as the evolution of hybrid inviability relies on genetic incompatibilities expressed in hybrids[1,42]. We tested for an interaction between isolating barrier (pre- or post-mating) and plasticity

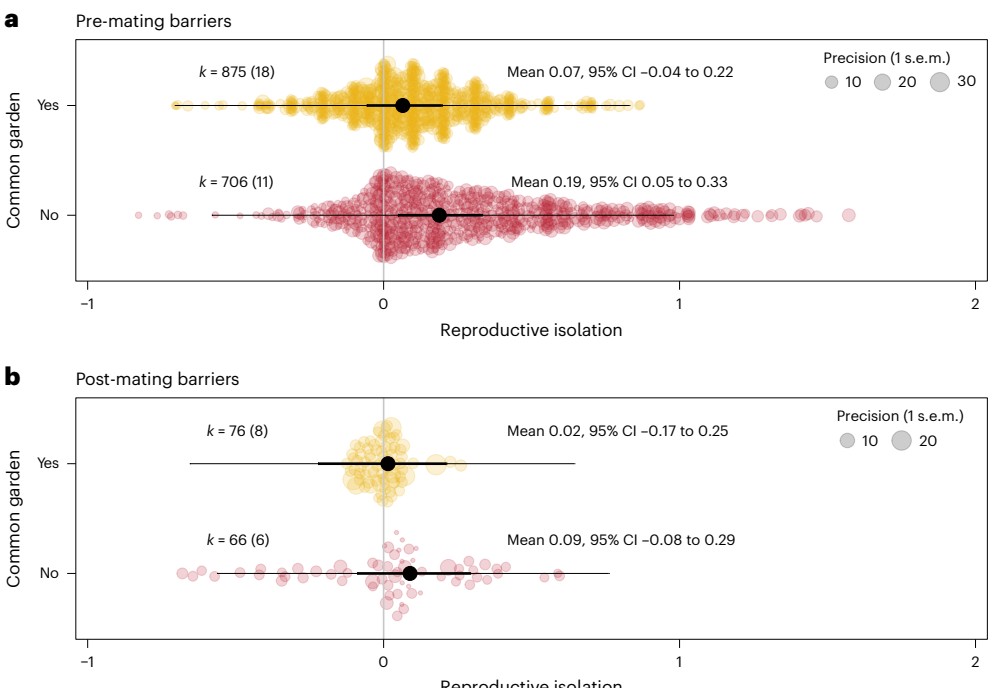

**Fig. 3 | Plasticity increases reproductive isolation, but only for pre-mating barriers. a**, Populations that were reared in a common garden environment (yellow) had lower estimates of pre-mating reproductive isolation than populations that did not pass through a common garden environment (red). **b**, In contrast, whether the population was reared in a common garden environment did not impact post-mating isolating barriers. Further, pre-mating isolation estimates (**a**) are significantly greater than post-mating isolation estimates (**b**). Reproductive isolation is *z*-transformed. *k* indicates the number of effects sizes extracted with the number of studies in parentheses. The thick black lines are the 95% credible intervals, and the thin black lines are the prediction intervals. Note that the precision legend is different in **a** and **b**.

by leveraging the variation across studies in their use of a common garden generation. Common garden designs typically involve rearing all populations in the same environment for at least one generation to remove effects of plasticity before assessing any evolutionary change that has occurred in the populations[28]. We asked whether reproductive isolation was greater when a common garden was not used to purge the environmental effects on phenotypes. We expected such an effect of plasticity only on pre-mating barriers but not on post-mating barriers. We indeed found that plasticity interacted significantly with the type of isolating barrier, and therefore plasticity probably plays an important role in promoting the early emergence of reproductive isolation but not necessarily in the later stages (Fig. 3). Specifically, populations that did not experience a common garden generation before the estimation of reproductive isolation exhibited significantly higher estimates of pre-mating reproductive isolation compared with populations that were exposed to a common garden environment (Fig. 3a), whereas estimates of post-mating isolation did not differ (difference estimate = −0.173, 95% CIs = [−0.346, −0.025], pMCMC = 0.034; Fig. 3b).

Phenotypic plasticity, including learned mate preferences, can promote pre-mating reproductive isolation. When reproductive isolation is linked to morphological traits, phenotypic plasticity could increase the phenotypic differentiation between populations over and above any genetic differentiation that may have evolved, thus increasing the degree to which the populations are reproductively isolated from one another. For example, diet, a common environment used in the studies here, can induce morphological shifts in body size that, in turn, can result in reproductive isolation between populations that developed in different environments[39]. In addition, environmentally induced differences in other morphological traits or chemical signals could accentuate underlying patterns of assortative mating that already exist within populations[43–46]. However, without a correlation between parental and offspring environments,

environmentally induced assortative mating will have little effect on speciation dynamics[37].

Learning is a special form of plasticity that can immediately impact the extent of gene flow between populations[47,48]. Offspring can imprint on their natal habitat[49,50], which, in combination with environmentally induced assortative mating, could promote reproductive isolation. Offspring can also imprint on their parents' phenotypes[51–53] or, in the case of brood parasitic birds, their host species[54,55]. Although environmental factors can promote reproductive isolation through various forms of plasticity, such environmentally induced reproductive isolation is likely to be fragile and collapse as soon as the environment changes. Indeed, species that are reproductively isolated through pre-mating phenotypic plasticity might be ephemeral[56–58]. Breakdown of ecological differences between recently diverged populations could then result in incomplete speciation[34] or speciation reversal[59], with species persisting if the plastic component of reproductive isolation is followed by genetic accommodation[60] or when intrinsic post-mating isolating barriers have subsequently evolved[61], which are not affected by environmental conditions (Fig. 3b).

### Evolution of isolating barriers mirrors macroevolution

Our study focuses on the onset of speciation, but are these findings also relevant to the end of speciation? The link between the microevolution of reproductive isolation and macroevolutionary speciation rates has been subject to much recent discussion and is far from resolved[5,44,62]. Comparative analyses suggest that pre-mating isolating barriers probably evolve earlier in the speciation process and are usually initially stronger than post-mating barriers[63,64], especially when taxa are sympatric, suggestive of reinforcement[17]. Consistent with this, and over and above the interaction with a common garden generation, we found that pre-mating reproductive isolation is stronger than post-mating reproductive isolation in experimental speciation studies

(difference estimate = 0.182, 95% CIs = [0.099, 0.272], pMCMC < 0.001; Fig. 3). However, post-mating isolating barriers can evolve just as fast, if not faster, than pre-mating barriers, at least in some taxa[46,65]. Our meta-analysis on experimental speciation studies has highlighted the dearth in post-mating estimates of reproductive isolation as most studies have focused on the consequences of mating (for example, decreases in longevity and fecundity). There are only a handful of estimates of hybrid fitness traits, which are often measured at the latter end of the speciation process. A consensus on the reproductive isolation components that can be measured both in the laboratory and in the field is therefore clearly needed to align studies that focus on the onset of speciation, and its completion.

## Conclusions

Here, we have presented meta-analytical evidence of experimental speciation studies that causally support the ecological speciation model for the evolution of reproductive isolation. Our meta-analysis provides quantitative evidence that (1) divergent selection can promote the rapid evolution of reproductive isolation (Fig. 2a), (2) phenotypic plasticity contributes to pre-mating reproductive isolation (Fig. 3) and (3) pre-mating isolating barriers evolve faster that post-mating barriers (Fig. 3). Our study also shows that the evolutionary dynamics of reproductive isolation in the early stages of speciation is partly decoupled from the dynamics further along in the speciation process (Fig. 2b). While recent work has suggested that ecological speciation may not be the most common mechanism by which species originate[15], our results suggest that divergent selection can rapidly produce populations that display significant reproductive isolation, potentially becoming incipient species, which is consistent with ecological speciation theory. Further, we show that plasticity could play a key role in initiating reproductive isolation, but its role in sustaining reproductive isolation may depend on the evolution of plasticity itself[60].

A few model organisms have dominated ecological speciation research: *Anolis* lizards, stickleback fish (*Gasterosteus aculeatus*), cichlid fish, *Timema* walking sticks, *Rhagoletis* flies and cynipid gall wasps. These model systems have all provided key insights into the process of speciation, but researchers have seldom quantified how reproductive isolation evolves in real time, across generational scales. The strength of the experimental speciation approach is that it has revealed the potential and importance of ecological speciation more generally and across a wider range of taxa. Future experimental speciation studies should broaden both the species used and the dimensionality of selection, while measuring several isolating barriers (both pre- and post-mating) at multiple timepoints to advance our understanding of the evolution of reproductive isolation in the early stages of speciation.

## Methods
### Overview
To quantitatively synthesize studies on the experimental evolution of reproductive isolation, we conducted a systematic literature search and meta-analysis. This involved five main steps: (1) defining study eligibility criteria, (2) building and validating a search string to find studies meeting these criteria, (3) conducting our search in Web of Science and Scopus and screening studies, (4) calculating effect sizes from our final sample of studies and (5) building statistical models to estimate mean effect sizes and to explore the effects of generation time and study design on estimates of reproductive isolation.

### Eligibility criteria
To be included in our dataset, studies had to meet the following criteria: (1) the study had to be in English; (2) the study is a peer-reviewed research paper; (3) the study needed to include data from live organisms from any taxon; (4) the study used experimental evolution with a founding population from a single genetic source (that is, not from multiple wild populations that may or may not have experienced different

selective pressures with its own genetic architectures that may or may not bias the path of evolution and subsequent reproductive isolation); (5) data had to be available, either as raw data or as means with associated standard errors/deviations and sample sizes; and (6) selection did not directly act on reproductive isolation itself (for example, studies were excluded if artificial selection was performed on assortative mating or on a putatively sexual trait). Our PICO components were population, experimentally-evolved living organisms; intervention, experimental evolution in different environments and with measures of reproductive isolation between populations (for example, mate choice including latency to mate or interact, habitat choice, fecundity, longevity after mating and hybrid fitness); comparison, estimates of reproductive isolation between populations evolving in the same environment compared with populations evolving in different environments; outcome, estimate of reproductive isolation acting via any isolating barrier (for example, pre- or post-mating barriers). From this, we constructed two decision trees, one for title and abstract screening and one for full-text screening (Supplementary Fig. 1).

### Search string development and validation
To build our search strings, we chose 15 benchmark studies that we wanted our literature search to return (Supplementary Table 1). These studies were chosen for their importance in testing the idea of divergent selection, sexual conflict and genetic bottlenecks in generating reproductive isolation using experimental evolution. The titles and abstracts from these studies were used to construct a word cloud (Supplementary Fig. 2), which enabled us to identify common terms and synonyms used across studies. We wanted our search to return between 1,000 and 3,000 studies. The final search strings (separated by (AND), Boolean operator 'AND') were

'divergent selection' AND hybrid (AND) 'experimental evolution' AND 'ecological speciation' (AND) 'experimental evolution' AND speciation (AND) 'experimental evolution' AND 'reproductive isolat*' (AND) 'hybrid inviab*' OR 'hybrid steril*' AND 'experimental evolution' (AND) (prezygotic OR postzygotic OR pre-zygotic OR post-zygotic) isolation AND 'experimental evolution' (AND) 'experimental evolution' AND 'sexual isolation' (AND) 'host plant' AND hybrid AND isolati* (AND) 'experimental evolution' AND adapt* AND divergen* AND speciation (AND) bottleneck AND 'reproductive isolat*' (AND) bottleneck AND speciation AND 'experimental evolution' (AND) 'founder flush' OR 'founder-flush'.

We validated the search string, before title and abstract screening all articles, by calculating its miss rate (the percentage of the benchmark papers missing) and hit rate (the percentage of papers that pass to full-text screening out of a random 100 papers). The miss rate was 13% (2/15 benchmark studies were missing) and the hit rate was 7% (of 100 random papers, 7 passed to full-text screening), which are typical in the field. Note, that we used our title and abstract decision tree (Supplementary Fig. 1) to calculate the hit rate.

### Searches and screening
We performed two topic searches using Web of Science and Scopus, adapting our search string for each. The first search was done in March 2021 (from Lund University, Sweden) to search for relevant studies, and the second in June 2023 (from Bangor University, UK) to account for studies published since our first search. The databases covered by Web of Science in each institution are reported in Supplementary Table 2. Only studies published in 1994 and beyond (after Rice and Hostert[10], published in December 1993) were considered.

In total, our literature search returned 2,715 studies (Supplementary Fig. 3). After removing 1,008 duplicates, we screened the titles and abstracts for 1,707 studies in Rayyan[66] following our decision tree. We selected 82 studies for full-text screening, to which we added

nine studies from other sources, which mainly included citations in screened papers[67–75] and five studies cited in Edelaar[76], a review published in 2022[77–81]. Of these 96 studies, 48 were included, with 34 studies[23,31,32,67,82–109] eligible for the meta-analysis concerned with the role of divergent selection in generating reproductive isolation (Supplementary Table 3).

## Data extraction and effect size calculation

All data manipulation and analyses were done in R[110]. Reproductive isolation can be viewed as an emergent property of the interaction between two populations. We therefore sought to include all estimates of reproductive isolation at the population level. For example, if a study had two replicate populations (1 and 2) evolving in two different environments (A and B), we would extract two within-treatment estimates of reproductive isolation (A1 × A2 and B1 × B2), and four between-treatment estimates of reproductive isolation (A1 × B1, A1 × B2, A2 × B1 and A2 × B2). As the same populations contribute to different estimates of reproductive isolation from the same paper, we accounted for this by calculating within-research group variance (co)variance matrices for each specific population used. To do this, we used the make_VCV_matrix function from the 'metaAidR' package[111]. In some cases, population-level estimates of reproductive isolation were not possible. This may be because the raw data were not made available to us or because population-level estimates were underpowered, as many papers pooled individuals from different populations to assess the broad effects of different environments on the evolution of reproductive isolation. Effect sizes that were estimates at the population–population level are coded as '1' in the 'pop_level_data' column (1,283/1,723 effect sizes), with '0' marking the estimates of reproductive isolation where pooling of replicate populations occurred at any stage.

To calculate effect sizes, we preferred to use the raw data provided in the Supplementary Information (1,42/1,723 effect sizes). Raw data were reanalysed using the 'brms' R package[112,113] using binomial, beta or Bernoulli distributions with logit link functions for mate choice data, a Gaussian distribution for relative fitness data or a gamma distribution with an exponential link function. For papers that did not provide supplementary data, we emailed the corresponding author to request it (98/1,723 effect sizes). In cases where authors did not respond, we extracted means and errors from figures using the R package 'metaDigitise'[114], randomly generating data with the same parameters using an appropriate distribution (normal, truncated normal and Poisson were used), and estimated reproductive isolation from these (83/1,723 effect sizes), a method used by Noble et al.[115]. Effect sizes calculated this way are coded as '1' in the 'backtransformed' column of the data.

**Metadata.** We collected metadata associated with our estimates of reproductive isolation. First, we grouped each estimate of reproductive isolation as to whether the isolating barrier operates 'pre-mating' (1,581/1,723 effect sizes) or 'post-mating' (142/1,723 effect sizes). Pre-mating barriers operate before mating and fertilization, and in our dataset include habitat selection (habitat choice or oviposition choice by females), with 184/1,581 effect sizes, and the most common form, sexual isolation (mate choice, latency to mate and mating duration) with 1,397/1,581 effect sizes. Post-mating barriers operate after mating and can either occur before or after fertilization. Post-mating barriers can be intrinsic, where genetic incompatibilities exist between the two populations, or extrinsic, where the barrier operates via genetic differences between population but in conjunction with some aspect of the environment. In experimental speciation experiments, post-mating estimates of reproductive isolation are measured much less frequently because they are more difficult to measure. Across the studies in our dataset, we classified post-mating isolation barriers into three broad categories: hybrid fitness (where hybrids are produced and fitness in any environment is measured, 34/142 effect sizes), fecundity (where

the number of eggs or offspring is measured for hybrid crosses, 49/142 effect sizes) and female longevity (where female longevity is measured after mating or harassment from males from a different population, 59/142).

Second, we collected metadata on whether reproductive isolation was estimated after every population had gone through a common garden generation or not. Common garden generations are used to remove effects of plasticity on trait values as all individuals experience the same developmental environment. Any differences in trait values or, in our case, reproductive isolation, could then be more confidently attributed to genetic changes in the population[30]. In all papers, this information was readily available.

Third, we noted the generation of the experiment at which reproductive isolation was estimated. This was our estimate of divergence between populations. The median number of generations at which reproductive isolation was estimated in our dataset is 30 generations, with a range of 8–1,589 generations. We centred the number of generations on 10 for inclusion in statistical models.

Last, the initial population size may influence the rate at which populations evolve reproductive isolation. In a similar vein to genetic bottlenecks (papers related to which were uncovered by our literature search but are not the subject of this study), smaller populations may be more prone to genetic drift, where different alleles may become randomly fixed in different populations, with these genetic differences being responsible for reproductive isolation. Alternatively, larger populations may be more likely to exhibit reproductive isolation as adaptation to divergent selection and the build-up of genetic differences between populations may be more efficient than in smaller populations. We obtained this information from either the paper that reported estimates of reproductive isolation or an earlier paper from the same research group that outlined the founding populations in more detail. We logarithmic-transformed the initial population size for inclusion in statistical models.

## Effect size summary

Invertebrates unsurprisingly dominate our dataset (Supplementary Table 3). *Drosophila* ($N_{study}$ = 11), *Callosobruchus maculatus* ($N_{study}$ = 4) and *Tetranychus urticae* ($N_{study}$ = 3) comprise the majority of the studies, with other invertebrate species filling out the majority of the dataset. Two fungi species (*Saccharomyces cerevisiae* and *Neurospora crassa*) and two vertebrate species (the domesticated chicken *Gallus gallus* and the Trinidadian guppy *Poecilia reticulata*) make up the final species (Fig. 1d).

## Statistical analyses

**Generation of the phylogenetic tree.** As multiple species are present in our dataset, we needed to control for phylogenetic signal that may be present in our dataset (that is, closely related species may respond to divergent selection more similarly than more distantly related species). We used the 'rotl' R package[116] to construct our cladogram, making use of the tnrs_match_names and tol_induced_subtree functions (Fig. 1d). We then used the Grafen method to compute the branch length using the 'ape' package[117].

**Publication bias and heterogeneity.** We explored the effects of publication bias using Egger's test and time-lag analysis[118]. Publication bias results from studies remaining unpublished (for example, due to nonsignificant findings), which can skew the distribution of effect sizes in a meta-analysis. Models investigating publication bias and heterogeneity are included in the related code and were performed both in 'MCMCglmm'[119] and 'metafor'[120] in R[110]. All priors and random effects are consistent between these models and the models used for the main analyses.

In our dataset, studies with smaller sample sizes reported larger effect sizes, indicating funnel asymmetry (Supplementary Fig. 4). The

slope of the relationship between our reproductive isolation effect sizes and their sampling variances was not significant (Egger's test: slope = 0.511, pMCMC = 0.03) in a multilevel meta-regression model accounting for repeated measures on different species and from different research groups, and non-independence due to phylogeny (Supplementary Table 4). The intercept of this model was 0.102 (pMCMC = 0.06), which provides an estimate of the adjusted mean effect size (that is, the mean amount of reproductive isolation when the sampling variance is zero).

There was no evidence of a time-lag bias, which occurs when stronger effects are published more rapidly than weaker ones, in a multilevel meta-regression model accounting for dependencies in the data (slope = 0.001, pMCMC = 0.84). A trim and fill analysis indicated that 0 (s.e.m. = 21.82) studies are needed to produce a symmetric funnel plot, and the fail-safe number ranged from 1,723 to >1 million, depending on the method of calculation. Note that the trim and fill analysis and the fail-safe numbers do not account for dependencies in the data. There was considerable heterogeneity in our dataset (Supplementary Table 4).

**Main analysis.** The main analyses were performed using both 'MCMCglmm'[119] and 'metafor'[120] in R[110]. Both results are qualitatively and quantitatively similar, so we only presented the results from 'MCMCglmm'. We used Inverse-Wishart priors for the random effects, and the default priors for fixed effects, with 1,020,000 iterations, 20,000 of which were burn ins, and a thinning interval of 1,000. Three chains were run for each model with different initial conditions, which were visually inspected for convergence as well as using Gelman's diagnostic test, which confirmed all chains mixed well and converged on the same results.

We first fit a full model that included the cross type (either within environments or between environments), the number of generations at which reproductive isolation was estimated (centred on ten generations), the barrier (pre- or post-mating), common garden (did the populations pass through a common garden, 'Yes' or 'No'), whether the data were at the population level ('Yes', or estimates combined multiple populations, 'No'), whether the data were backtransformed ('Yes' or 'No') and the founding population size of the populations used in the experiment (logarithmic-transformed).

We tested for three different interactions in three separate models, each including all the covariates listed above: (1) the interaction between time and cross type, which was not significant (see 'Reproductive isolation does not increase with time'); (2) the interaction between cross type and whether a common garden generation was used, which was not significant (estimate = 0.026, 95% CIs = [−0.04, 0.10], pMCMC = 0.49); and (3) the interaction between the barrier type (pre- or post-mating) and whether or not a common garden generation was used, which was significant (see 'Plasticity increases pre-mating reproductive isolation'). As there was support for the third interaction, we report estimates from this model.

We included species, the species phylogeny and the research group in which the work was completed as random terms. We included research group instead of paper ID as four studies measured reproductive isolation on two experimental evolution experiments (see Supplementary Table 3 for more information). As we used multiple estimates of reproductive isolation from the same populations within experimental evolution studies, we used a variance (co)variance matrix of sampling variances that accounted for covariance between populations as the sampling variances for the model. The analysis using 'metafor' matched the results using 'MCMCglmm' presented in the manuscript, but we display the 'metafor' results in Supplementary Table 5.

**Studies where reproductive isolation is estimated more than twice.** We found no effect of the number of generations a population evolved on reproductive isolation with the complete dataset. This is

an unexpected result, which could be a consequence of comparing species with substantially different generation times (for example, the domestic chicken versus *Drosophila*), being sampled at different timepoints. For example, if species with long generation times were sampled after relatively few generations, there may be limited reproductive isolation, which would pull the overall slope towards zero. To more explicitly test this, we conducted a within-species analysis, focusing on only the studies that estimated reproductive isolation between the same populations more than twice. This allowed us to estimate the effect of generation time on reproductive isolation for five different species (seven papers from five research groups; Supplementary Fig. 5).

For each species, we ran a simple linear model with zRI as the response variable and generation time as the fixed effect. We then combined the intercepts and slopes from these five linear models in a meta-analytic model, which incorporates the variances and covariances of the sampling errors of these parameters (known as a two-stage analysis in the 'metafor' R package). This gave us an estimate of the mean slope and intercept across species. We found that the mean intercept was equal to 0 (estimate = 0.73 ± 0.39 s.e.m., 95% CIs = [−0.17, 1.64], $t = 1.88$, $P = 0.09$), but suggestive that reproductive isolation evolved at the onset of speciation in this subset of the data. We did not find evidence that the mean slope estimate was positive or indeed different from 0 (slope estimate = −0.17 ± 0.12 s.e.m., 95% CIs = [−0.45, 0.11], $t = −1.41$, $P = 0.20$). The purpose of this analysis was to focus on the evolution of reproductive isolation through time (generations) and so we did not investigate differences in slope for estimates of reproductive isolation between and within environments. This is not possible given the available data. This drastically reduced dataset also prevented a phylogenetically informed approach.

We confirmed these findings using a random intercept and slope model in 'MCMCglmm' with the same dataset of five research groups. This fits a random slope and intercept for each research group. The intercept estimate was higher in this model than the two-stage model, although it was not statistically different from 0 (estimate = 0.85, 95% CIs = [−0.46, 2.01], pMCMC = 0.15). As above, this analysis also found no evidence that the number of generations changed the estimate of reproductive isolation (slope estimate = −0.21, 95% CIs = [−1.04, 0.48], pMCMC = 0.54).

## Sensitivity analysis
In this section, we will outline a range of analyses to assess the robustness of the dataset. In many case, small decisions about which effect sizes to include, or which set of errors to include, may influence the overall results. This section is designed to explore some of these small decisions and their impact on our main results.

**Use of $1/(n-3)$ sampling variance.** For Fisher's $z$-transformed data of correlation coefficients, as our estimates of reproductive isolation effectively are being bounded between −1 and +1, $1/n-3$ is used as an estimate for the sampling variance for each effect size, where $n$ is the sample size. For our main analyses, we used the sampling variance obtained by squaring the standard deviation of the posterior distribution of the transformed estimate. For completeness, we outline the main results using the sampling variance of $1/(n-3)$.

The use of $1/(n-3)$ does not change the main results. We find that (1) reproductive isolation estimates between environments is greater than reproductive isolation within environments (difference estimate = 0.07, 95% CIs = [0.05, 0.11], pMCMC < 0.001), (2) pre-mating reproductive isolation estimates are greater than post-mating estimates of reproductive isolation (difference estimate = 0.18, 95% CIs = [0.09, 0.26], pMCMC < 0.001), (3) a common garden generation interacts with the isolating barrier (difference estimate = −0.17, 95% CIs = [−0.32, −0.02], pMCMC = 0.024) and (4) the number of generations (slope estimate = $−2.96 \times 10^{-6}$, 95% CIs = [$−1.24 \times 10^{-4}$, $1.48 \times 10^{-4}$], pMCMC = 0.96), the founding population size (slope estimate = 0.004, 95% CIs = [−0.01, 0.02, pMCMC = 0.65) and whether the data were

'backtransformed' (difference estimate = −0.10, 95% CIs = [−0.21, 0.01], pMCMC = 0.06) or from population-level crosses (difference estimate = 0.05, 95% CIs = [−0.02, 0.10], pMCMC = 0.16) did not influence the evolution of reproductive isolation.

**Exclusion of within-environment habitat isolation.** Our metric of reproductive isolation varies between −1 and +1 except for within-environment estimates of habitat isolation. In this case, the metric of isolation is on which host plant females choose to oviposit. If two populations have been evolving on the same host plant and both choose to oviposit on that same host plant, reproductive isolation will equal 0. Estimates of reproductive isolation cannot go below 0. We therefore excluded these data in the analysis below and reanalysed the dataset. There are $N = 78$ estimates of within-environment habitat isolation that are excluded from this analysis.

Our results still stand for this analysis: reproductive isolation estimates between environments is greater than reproductive isolation estimates within environments (difference estimate = 0.09, 95% CIs = [0.05, 0.12], pMCMC < 0.001); pre-mating reproductive isolation estimates are greater than post-mating reproductive isolation estimates (difference estimate = 0.18, 95% CIs = [0.08, 0.27], pMCMC < 0.001); plasticity increases pre-mating isolation, but not post-mating isolation (difference estimate = −0.16, 95% CIs = [−0.30, −0.01], pMCMC = 0.024); the number of generations does not influence reproductive isolation (slope estimate = $-2.78 \times 10^{-5}$, 95% CIs = [$-1.56 \times 10^{-4}$, $2.26 \times 10^{-4}$], pMCMC = 0.848); and the founding population size does not influence reproductive isolation (slope estimate = 0.003, 95% CIs = [−0.01, 0.02], pMCMC = 0.73). Whether the data are 'backtransformed' does influence reproductive isolation (difference estimate = −0.12, 95% CIs = [−0.20, −0.01], pMCMC = 0.096), contrary to the main results.

### Reporting summary

Further information on research design is available in the Nature Portfolio Reporting Summary linked to this article.

## Data availability

All data associated with this paper are available via figshare at https://doi.org/10.6084/m9.figshare.27233727 (ref. 121).

## Code availability

All code associated with this paper are available via figshare at https://doi.org/10.6084/m9.figshare.27233727 (ref. 121).

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

## Acknowledgements

We acknowledge a Human Frontiers Science Program long-term fellowship (LT000879/2020) for funding B.J.M.J., a Marie Skłodowska-Curie Postdoctoral Fellowship (project number 101067861) for funding P.A.D. and the Swedish Research Council for funding E.I.S. (VR: 2020-03123). We thank M. Björklund, A. Chippindale, S. Magalhães, N. G. Prasad, H. Rundle, V. Shenoi and H. Thyagarajan for kindly provided requested data, and the permission to upload the data with this manuscript. We also thank A. Comeault, S. Nilén, D. Parker, I. Prates and M. Tsuboi for comments.

## Author contributions

B.J.M.J. and E.I.S. conceived the study, B.J.M.J. collected the data, B.J.M.J. and P.A.D. analysed the data, B.J.M.J. wrote the first draft, and all authors contributed to the final version.

## Competing interests

The authors declare no competing interests.

## Additional information

**Correspondence and requests for materials** should be addressed to Benjamin J. M. Jarrett.

# Reporting Summary

## Statistics

For all statistical analyses, confirm that the following items are present in the figure legend, table legend, main text, or Methods section.

| n/a | Confirmed | |
|---|---|---|
| ☐ | ☒ | The exact sample size ($n$) for each experimental group/condition, given as a discrete number and unit of measurement |
| ☐ | ☒ | A statement on whether measurements were taken from distinct samples or whether the same sample was measured repeatedly |
| ☐ | ☒ | The statistical test(s) used AND whether they are one- or two-sided *Only common tests should be described solely by name; describe more complex techniques in the Methods section.* |
| ☐ | ☒ | A description of all covariates tested |
| ☐ | ☒ | A description of any assumptions or corrections, such as tests of normality and adjustment for multiple comparisons |
| ☐ | ☒ | A full description of the statistical parameters including central tendency (e.g. means) or other basic estimates (e.g. regression coefficient) AND variation (e.g. standard deviation) or associated estimates of uncertainty (e.g. confidence intervals) |
| ☐ | ☒ | For null hypothesis testing, the test statistic (e.g. $F$, $t$, $r$) with confidence intervals, effect sizes, degrees of freedom and $P$ value noted *Give P values as exact values whenever suitable.* |
| ☐ | ☒ | For Bayesian analysis, information on the choice of priors and Markov chain Monte Carlo settings |
| ☐ | ☒ | For hierarchical and complex designs, identification of the appropriate level for tests and full reporting of outcomes |
| ☐ | ☒ | Estimates of effect sizes (e.g. Cohen's $d$, Pearson's $r$), indicating how they were calculated |

*Our web collection on statistics for biologists contains articles on many of the points above.*

## Software and code

Policy information about availability of computer code

| Data collection | Data were extracted from papers that successfully passed through the PRISMA checklist. Raw data was obtained from the supplementary information, from the authors, or means and errors were extracted from the images/tables using metaDigitise_1.0.1 |
|---|---|
| Data analysis | R (4.2.2) was used for all analyses. Packages used for analysis were: metaAidR_0.0.0.9000, MCMCglmm_2.35, metafor_4.4.0, brms_2.18.0 |

For manuscripts utilizing custom algorithms or software that are central to the research but not yet described in published literature, software must be made available to editors and reviewers. We strongly encourage code deposition in a community repository (e.g. GitHub). See the Nature Portfolio guidelines for submitting code & software for further information.

## Data

Policy information about availability of data

All manuscripts must include a data availability statement. This statement should provide the following information, where applicable:

- Accession codes, unique identifiers, or web links for publicly available datasets
- A description of any restrictions on data availability
- For clinical datasets or third party data, please ensure that the statement adheres to our policy

Data are available as supplementary information and on figshare (https://doi.org/10.6084/m9.figshare.27233727)

# Research involving human participants, their data, or biological material

Policy information about studies with human participants or human data. See also policy information about sex, gender (identity/presentation), and sexual orientation and race, ethnicity and racism.

| Reporting on sex and gender | NA |
|---|---|
| Reporting on race, ethnicity, or other socially relevant groupings | NA |
| Population characteristics | NA |
| Recruitment | NA |
| Ethics oversight | NA |

Note that full information on the approval of the study protocol must also be provided in the manuscript.

# Field-specific reporting

Please select the one below that is the best fit for your research. If you are not sure, read the appropriate sections before making your selection.

☐ Life sciences          ☐ Behavioural & social sciences          ☒ Ecological, evolutionary & environmental sciences

For a reference copy of the document with all sections, see nature.com/documents/nr-reporting-summary-flat.pdf

# Ecological, evolutionary & environmental sciences study design

All studies must disclose on these points even when the disclosure is negative.

| Study description | The study is a meta-analysis of estimates of reproductive isolation obtained in experimental speciation studies. A single population is split into replicates that experience the same (within treatment) or divergent (between treatment) environments. Reproductive isolation was estimated between populations within and between treatments to quantitatively assess ecological speciation at its onset. |
|---|---|
| Research sample | Data were obtained from papers searched for using a search string and filtered using a PRISMA flow diagram. The dataset compiled was 1723 effect sizes from 34 studies on 15 species. |
| Sampling strategy | Any relevant paper found in the two searches were kept for inclusion. If no estimate of reproductive isolation could be calculated (i.e., if there was no raw data or means with errors reported) the studies were excluded. |
| Data collection | One author, BJMJ, extracted all the data and calculated the appropriate effect sizes. |
| Timing and spatial scale | The search spanned the publication of Rice and Hostert (1993) in December 1993 to June 2023, when the final search was performed. |
| Data exclusions | No relevant data were excluded. |
| Reproducibility | All the code to extract the raw data from the filtered studies, as well as the code for the final meta-analysis, is included in the supplementary material. |
| Randomization | There was no randomisation as we only had the data that was available to us, either from means and standard errors published in the paper or via the raw data. |
| Blinding | Blinding was not possible for this study. BJMJ worked on extracting the data from each paper in turn. |

Did the study involve field work?          ☐ Yes          ☒ No

# Reporting for specific materials, systems and methods

We require information from authors about some types of materials, experimental systems and methods used in many studies. Here, indicate whether each material, system or method listed is relevant to your study. If you are not sure if a list item applies to your research, read the appropriate section before selecting a response.

## Materials & experimental systems

| n/a | Involved in the study |
|-----|----------------------|
| ☒ | ☐ Antibodies |
| ☒ | ☐ Eukaryotic cell lines |
| ☒ | ☐ Palaeontology and archaeology |
| ☒ | ☐ Animals and other organisms |
| ☒ | ☐ Clinical data |
| ☒ | ☐ Dual use research of concern |
| ☒ | ☐ Plants |

## Methods

| n/a | Involved in the study |
|-----|----------------------|
| ☒ | ☐ ChIP-seq |
| ☒ | ☐ Flow cytometry |
| ☒ | ☐ MRI-based neuroimaging |

## Plants

| | |
|---|---|
| Seed stocks | NA |
| Novel plant genotypes | NA |
| Authentication | NA |

