## [Peer Review File · Nature Ecology & Evolution]

Meta-analysis reveals that phenotypic plasticity and divergent selection promote reproductive isolation during incipient speciation

Corresponding Author: Dr Benjamin Jarrett

Version 0:

Decision Letter:

23rd December 2024

Dear Ben,

Your manuscript entitled "Divergent selection and phenotypic plasticity promote reproductive isolation" has now been seen by 3 reviewers, whose comments are attached. The reviewers have raised a number of concerns which will need to be addressed before we can offer publication in Nature Ecology & Evolution. We will therefore need to see your responses to the criticisms raised, along with a revised manuscript, before we can reach a final decision regarding publication.

While all referees find the study interesting, they have each requested clarifications for the hypothesis, methods, and interpretation of results. Reviewers 1 and 3 in particular have suggested additional analyses. We therefore invite you to revise your manuscript taking into account all reviewer comments. Please highlight all changes in the manuscript text file.

* If you have not done so already please begin to revise your manuscript so that it conforms to our Article format instructions at <http://www.nature.com/natecolevol/info/final-submission>. Refer also to any guidelines provided in this letter.

Link Redacted

To improve transparency in authorship, we request that all authors identified as 'corresponding author' on published papers create and link their Open Researcher and Contributor Identifier (ORCID) with their account on the Manuscript Tracking System (MTS), prior to acceptance. ORCID helps the scientific community achieve unambiguous attribution of all scholarly

contributions. You can create and link your ORCID from the home page of the MTS by clicking on 'Modify my Springer Nature account'. For more information please visit www.springernature.com/orcid.

We look forward to seeing the revised manuscript and thank you for the opportunity to review your work. I hope you will have a great holiday season and happy new year.

[redacted]

Reviewer expertise:

Reviewer #1: plasticity, meta-analysis

Reviewer #2: plasticity, speciation

Reviewer #3: plasticity, speciation theory

Reviewers' comments:

Reviewer #1 (Remarks to the Author):

I was excited to read this paper. It tackles a fundamental question in evolutionary biology, how does reproductive isolation evolve in the early stages of speciation? This is a crucial question for understanding patterns of speciation and remains a challenging question to completely nail down given that the speciation process is difficult to capture. The authors of this paper set up a creative and innovative way of testing these ideas by meta-analysing experimental evolution experiments to capture the early stages of reproductive isolation. Unsurprisingly, the analysis is mainly restricted to insects and a couple vertebrate systems which are the only systems that are tractable for such experiments. Nonetheless, this shouldn't detract from the important insights we gain from synthesising such empirical work.

The authors beautifully establish the working theory and clearly connect the effect size and moderators in the meta-analysis to that theory. The analyses are robust. I have reviewed a lot of meta-analysis in my career, and I can tell you that this one ranks highly. The systematic searches are done robustly, careful thought has gone into the effect size calculation and its meaning, sensitivity of results carefully explored, and the conclusions justified. While there is still some room for improvement, I'm convinced the findings will remain robust and make a fundamental contribution to the field with broad implications for understanding the speciation process. I provide more detailed comments below. While my comments seem substantial, I think they are all addressable. I hope these will help further improve what I think is an excellent contribution.

Comments:

L28-29. Something reads wrong with "This views..." sentence. It sounds awkward as written. May be cut and reword to: "A long-standing explanation for the evolution of reproductive isolation is the ecological speciation model^{4,7,14}, whereby reproductive isolation evolves as a by-product of natural selection operating in different environments^{4,14}"

L30-33. This is a mouthful. I'd suggest breaking this sentence into two. There are too many ideas here.

L37-40. Interesting hypothesis. I take it that this is dependent on levels of gene flow and population size? This isn't really mentioned. Small effective population sizes would result in genetic drift dominating which would fix different mutations in each population by chance, that makes sense to me. But wouldn't this also make selection less effective? The statements around this are unclear. It is almost implied that selection is important when stating: "also driven by selection", but then how can this be if you assume selection pressures are the same (as stated on L38)? As it stands, it's a little vague on the exact processes and how this plays out. I think some more detail on this hypothesis (which I'll admit I am not familiar with) is warranted. I suggest some rewording and adding some text to improve clarity.

L40-44. I think some more detail on this study is needed because everything stated here is a little vague. Its meaning is unclear without a bit more context around the study.

Fig 1B. Overall, figure 1 is a nice figure but I struggled to understand what was trying to be show in the panels of Fig 1B. The text was more useful but there needs to be more explanation of the 'grid' and what A and B are referring to, why are some boxes blank? I think there is clarity on L577-580, but this should be detailed in the legend as the figure should stand on its own.

L96. 'in' for 'n'

L158-159. I think this is the context that is needed in the methods to better set up the reasoning behind analyses. I suggest repeating briefly in the methods. (L710)

L164-166. Probably also worth pointing out that the slope estimate is negative, even if not significant, and with a reduced dataset you lose power, with this subset I think, based on what you state on L158-159, your prediction would actually be a negative slope, right? If so, pointing out that this slope is actually going in the predicted direction is important.

L179-180. I find it challenging to understand this point fully. I am not doubting that plasticity can drive phenotypic differences, but I'm surprised at the notion that such effects can drive reproductive isolation in a general sense as implied here. I can understand this for pre-mating isolation mechanisms but not post-mating. Can the authors expand on the references with a couple more sentences outlining the mechanism in a little more detail? Are there empirical examples where post-mating

isolation mechanisms are involved? Or do they really mean with respect to pre-mating isolation mechanisms and not post-mating? That would make more sense to me, but then, it should be clarified here. Is there sufficient data to test the hypothesis that this is only for pre-mating isolation mechanisms by adding an interaction between the isolation barrier moderator and whether it was a common garden? That would test this hypothesis directly. It may show me that I'm wrong! I realise, however, that there may not be enough data to test this interaction, but it would be an interesting addition because the intuition that this applies to pre- and post-mating isolation mechanisms may not be clear to readers. On L205-213 the authors argumentation seems to support my point as much of what is discussed seems to be pre-mating.

L234. I wouldn't be hesitant to suggest causality. I would suggest simply removing it. It's not needed in any case.

Fig 2a, 3, 4. Can you plot the line at 0 above the points? The yellow/red dots are covering it making it hard to match with the 95% CIs. Alternatively (and possibly in addition), adding some text with the mean effect and 95% CI would be good as it would provide the exact mean estimates and uncertainty. I'm aware that these are in the text, but I like these on the figures personally because there is no ambiguity.

L507-508. Inclusion criteria 2 is awkwardly worded. Do you mean "live" organisms and "any taxa" rather than "taxon unspecific".

L512-514. How was criteria 6 evaluated practically? I'd be clear on that because it may be that some studies didn't measure and present selection gradients. Would that be true?

L516. What were your measures of reproductive isolation? Be specific here. ON that note, this should also be detailed more on L518; For example, "via any isolating barrier", what do you mean specifically by this?

L547. Change "out" -> 'our'

L561-600. This metric makes sense, and I agree with what the authors have done, but I find the workflow odd. How they have written the description of the results creates a lot of conflicting information and leads to uncertainty about what was done and why. It wasn't until reading section 7.5.1 that I realised the authors are actually aware of the utility of the Bayesian approach they are taking, and this section was re-assuring to me because it shows the results were the same. I say this because from my experience in applying the same types of approaches the authors are taking here, sampling variance computed from the Bayesian methods versus those calculated from the sampling variance equations (formula for Z_r) can be quite different. In essence, the transformation from the effect size in each individual study script to Z_r could be done with the whole posterior distribution, and then, using that, the authors can take the SD of the posterior distribution of the effect size (which is the sampling standard error). Then, square it to get the sampling variance. Now you have both the effect size AND the sampling error directly from your Bayesian models. This is what section 7.5.1 is doing but this important point does not come across. I'd trust the Bayesian sampling variances more than the Z_r equation because the models used to calculate effects are binomial, beta etc. What I find odd in how this is written and presented is that, despite being aware of this (as suggested in 7.5.1) the authors opt to calculate the sampling error according to the typical sampling error for Z_r [i.e., use the point estimate from the Bayesian models then use "z_transform" AND $1/(n-3)$ to calculate the effect and its sampling error] as indicated in the first part of script "0. Overview.R". Why? It's good that these two methods say the same thing, but it does create a lot of confusion and defeats the purpose of using the raw data in the brms analyses. My suggestion would be to present only the results based on the sampling variances taken from the posterior distributions of each effect size in the main MS. Then, run the meta-analysis on that. It's more logically coherent with the Bayesian analyses the authors are using, is more accurate, and it means that the writing can remain focused and clear. I think it's more strongly justified as well. I'm assured given section 7.5.1 that this won't change the overall conclusions, but it will make the paper much stronger in presentation and will be analytically more coherent.

L597-600. Great way to creatively use the means and SDs to generate analogous distributions to your Bayesian ones!

We've used a very similar approach (Noble et al. 2019. PNAS. 116, 13452–13461; Radersma et al. 72. Evolution Letters 4, 360–370) using the multivariate normal. I wonder if the authors could just add a sentence to describe what distribution(s) was used for simulation? Beta-binomial? Poisson?

L672-675. Great to be thorough on the publication bias but many of the tools used are not appropriate, as pointed out by the authors (L674). My suggestion would be to stay focused on the regression approaches that include as moderators time lag and sampling variance / effective sample size. Report and correct effects on these. This section and details on 660-663 can simply be removed or just moved to supplement. Saves you space. It's important that the slope is presented from your multi-level meta-regression, but I'd also suggest adding the corrected effect size because the intercept from such a model corrects the overall effect size (technically) for publication bias.

L681-682. Remove the heterogeneity from the random effects model. It's not the appropriate model for these data.

L689. Did you run only a single chain? I'm fine with this if the chains are mixing well and they are run long enough, but I'd suggest adding some justification here as many folks using Bayesian estimation procedures insist on multiple chains.

Multiple chains can be useful and important to ensure that chains are converging to same answer as the MCMC process is inherently stochastic in its starting values and the ways in which it explores parameter space. I'd also like to be re-assured that chains are converging because when using the posterior distribution for the sampling variance it could matter.

L692. "full model" instead of "grand model" is more commonly applied terminology?

L687-708. I'd suggest making changes to the presentation and reporting of mean effects from this model to make them more easily interpretable and meaningful. This would also require some updates to the figures (sorry), but I think it will be worth it. It doesn't appear that you centred generation time on a biologically sensible number of generations, say, 10? If not (appears not) I'd suggest doing that because it will make your intercepts more meaningful. Alternatively, you can also calculate conditional mean estimates for different generation times for each of your categorical moderators (e.g., common garden, isolating mechanism etc) which would make your mean estimates in Fig 3 and 4 also easier to interpret. We discuss some thoughts on this in Noble et al. 2022. J Experimental Biology. 225, jeb243225 in a slightly different context but one that is nearly equivalent. The idea is that, if you centre generation time on something meaningful then you can be more precise about what the mean effect sizes are in your figures (2, 3, 4). In other words, they would be the mean effect size after, say, 10 generations. In fact, I would recommend these figures present multiple means at different generation times (i.e., condition the means on generation time) to demonstrate how means change through time. It would also imply that you can simply remove Fig 2 B because it would be wrapped up in the means at different generation times anyway. To be clear, I suggest having a

look at this paper O'Dea et al. 2019. *Fish and Fisheries*, 20: 1005-1022 (fig 6.) and also we show how to use orchard package to do that here: <https://daniel1noble.github.io/orchaRd/#example-6-meta-regression-and-conditional-mean-effect-sizes>.

L710 onwards: Sorry, I'm struggling to understand what the authors did here. It's unclear why subsetting the data to only studies estimating RI more than twice is useful from these few introductory sentences. I interpret it being useful because the authors have some studies that provide an effect size at more than one time point across generations. For example, for a study that does, say 30 generations, they measure RI, after say, 2 generations then test RI (giving you one effect size), then let them evolve for another 2 generations and then again measure RI. Giving you a second effect size. Is this the right way to think about it? If so, I'd suggest writing some of the logic / justification to explain this better. Maybe a clearer topic sentence stating the objective more clearly will help: "To understand XX we did YY". I think some of these points are better discussed in the main body of text, but it should come across down here too. I'm also unclear on what you mean by 'two-stage' model. It seems to me you fit a model on this subset with the structure: effect size ~ generation time. Is that true? If so, then doesn't the intercept just tell you what RI is at generation time 0 (i.e., just the start)? I'm sure this is not what the authors mean but some clarity about the model structure here is needed. Also, why the difference between MCMCglmm and metafor? Did you fit a random slope model in metafor? It's not the same as what MCMCglmm is doing so that could explain the difference. Either way, I think some more work on this section is needed to clarify the goals, predictions and the model structure. That will make the significance of the sections result become clearer to the reader.

L717: "The" rather than "Our"

L760-762. Again, I'd suggest the authors give some rationale about what the problem is to justify these analytical choices. From the simple topic sentences here, it's not clear why they even need to conduct these additional analyses. What's the problem they are trying to circumvent? Their main analyses are robust, and they seem to be concerned about running additional analyses that (I would argue) are less correct and thus likely to show conflicting results. It's unclear what they are trying to achieve in such analyses so at the very least some stronger justification is required.

Table S5. I can see from this table the authors are using default metafor options. In this case I don't think it will matter, but I would personally suggest using `test = "t"` so that the default test statistics is using the t-statistic and not the z-statistic. In addition, considering some simulations we have run (Nakagawa et al. 2022. *Ecology* 103, e03490) I'd also suggest using the `dfs = "contain"` argument. Wolfgang implemented this after we did these simulations and will correct the degrees of freedom to be more sensible when you have a lot of within study effects. This should also converge more to line up with your Bayesian models.

Hopefully my comments are useful. If anything is unclear do not hesitate to get in touch.

Daniel Noble

Reviewer #1 (Remarks on code availability):

Yes, the code has a README detailing the files and the data. Repository could be better organised so that all R script associated with each paper to extract the raw data is found within the PaperData/ folder and inside each papers folder. Otherwise, it's a little chaotic with so many files. Not totally bad because it can be reproduced still but it is a little overwhelming. I suggest the authors use `pacman::p_load()` function for installation of packages as code will fail if authors have not installed libraries. They should have a single call to `install.packages("pacman"); library(pacman)`, then they can list the packages.

Reviewer #2 (Remarks to the Author):

This manuscript presents the results of a meta-analysis in which the authors examined the role of divergent natural selection and phenotypic plasticity on the evolution of reproductive isolation. The authors' principal findings are: 1) the evolution of reproductive isolation by divergent selection is common and detectable across a range of taxa and environments, and 2) reproductive isolation does not appear to increase with time in experimental speciation studies. The first finding is consistent with ecological speciation theory. The second finding was unexpected, as it has long been assumed that reproductive isolation should increase with how long two populations have separated. The authors evaluate four alternative hypotheses to explain this surprising result, concluding that it reflects the action of phenotypic plasticity. Specifically, they suggest that divergent environments trigger a plastic increase in reproductive isolation in the first few generations.

Overall, I enjoyed reading this paper and have no major concerns. The data from the meta-analysis alone makes this paper worthwhile. I also appreciated the authors providing suggestions for future research at the end of their paper.

However, I have a few minor suggestions for the authors to consider. They are listed below in the order in which they appeared in the paper.

1) The first line of the Abstract should probably be rewritten. I suspect that many speciation researchers might disagree with the authors' claim that "the factors that shape its [reproductive isolation's] evolution during the early stages of speciation remain largely unknown." How about saying something like "the factors ... speciation require further clarification"?

2) Given the nature of this paper, it would be helpful to provide a general definition of reproductive isolation right up front. If there are slightly different definitions, the authors could provide a table listing some commonly used definitions.

3) On page 1, line 31, the authors introduce the idea that adaptation in response to divergent selection sets the stage for the evolution of reproductive isolation by resulting in genetic divergence between populations experiencing contrasting selection pressures. This claim is well supported theoretically and empirically, of course. Still, I wonder if this statement could be made more general by noting that divergent selection can result in divergence between populations IN FACTORS THAT CAN BE INHERITED. This would make the scenario outlined here more general, including cases in which reproductive isolation is mediated via learning or the transmission of symbionts.

4) On page 2, line 40, I recommend providing a quick synopsis of mutation-order speciation.

5) Page 3, line 96: typo: "n" should be "in" (as IN Rice and Hostert)

6) Page 4, lines 111-112, I could not find any explanation in the Methods regarding how the authors defined "same" versus "different" environments. Was this left up to the researchers of the original studies? Was there perhaps an effect on the evolution of reproductive isolation in how different the two environments were to each other that the authors of this study could analyze? This seems to be an important question that a meta-analysis like this could resolve.

7) Page 4, line 122: "genetic" => "inherited" (see point #3 above)

8) Page 5, line 157: The authors note that the experiments they reviewed for their meta-analysis imposed "a large magnitude of selection" on the experimental subjects. Can this magnitude of selection be quantified (or a range of values reported) to justify this claim?

9) Page 5, line 179: I would use the term "phenotypic plasticity" (rather than "developmental plasticity") throughout the paper since the former is a more widely used and general term. This paper does not need to distinguish between these two terms.

Overall, this is a nice manuscript that I hope to see in print soon!

Reviewer #3 (Remarks to the Author):

This study uses a meta-analytic approach to investigate the role of divergent selection in distinct environments in the evolution of reproductive isolation. This is a very interesting question. It's widely assumed that reproductive isolation evolves passively through accumulation of mutations that induce incompatibility via pleiotropic effects on reproductive traits. However, ecological speciation theory suggests that divergent selection can accelerate this process. The relative importance of these processes is not clear. This study addresses this question directly, and provides some novel insights. Nonetheless, I have a few suggestions and would like to see some additional analysis to clarify the findings.

To me, the most interesting finding was that plasticity seems to contribute a great deal to reproductive isolation. Indeed, the "difference estimate" for the role of plasticity (~ -0.1) was considerably greater than the "difference estimate" for the comparison of populations evolving in different vs. same environments (~ -0.07). This left me wondering whether there's any evidence that divergent selection contributed to isolation in these experiments, once the effect of plasticity is factored in. The data shown in Fig. 3 suggest that there is not: the mean effect size for studies that incorporated a common-garden step prior to isolation assays does not appear to be different from zero. It's thus potentially misleading to conclude that divergent selection promotes isolation based on an analysis that ignores the role of plasticity. To clarify this, it would be helpful to report separate analyses for studies that subjected their experimental animals to a common-garden step prior to isolation assays vs. studies that did not do so. Better yet, perhaps a model incorporating both environment (same vs. different) and plasticity (presence vs. absence of a common-garden step) could be used and the interaction of environment x plasticity could be tested. If this analysis shows that there's no evidence that divergent selection promotes isolation once the role of plasticity is factored in, that would be a very interesting finding!

Other comments:

L17: Provide a median and range for the number of generations here.

L28: This is an odd word to use here. Perhaps change to "posits" or "proposes"?

L53: What is a "speciation continuum"?

L96: Typo ("as n Rice and Hostert")

For each test, it would be helpful to state the number of effect sizes in the text.

L202-203: I wasn't sure what point the authors were aiming to make here. If you're thinking about populations inhabiting different environments that impose differential selection, then surely there will be consistent differences between those environments over many generations. Are you dismissing the possibility of sympatric speciation initiated by development in distinct habitat patches?

Figure 1: Add sample sizes for each taxon to panel D.

Looking at distributions of effect sizes in Figures 2, 3 and 4, there appear to be large differences in both the number of effect

sizes and their variance between the groups being compared. For example, the number of effect sizes for within vs. between-environment assays is very different, as is the variance of effect sizes. Does this require a correction for unequal variances, or is such a correction already incorporated into the analysis?

*****END*****

Version 1:

Decision Letter:

6th February 2025

Dear Ben,

Thank you for submitting your revised manuscript "Divergent selection and phenotypic plasticity promote reproductive isolation at the onset of speciation" (NATECOLEVOL-24102951A). It has now been seen again by the original reviewers and their comments are below. The reviewers find that the paper has improved in revision, and therefore we'll be happy in principle to publish it in Nature Ecology & Evolution, pending minor revisions to satisfy the reviewers' final requests and to comply with our editorial and formatting guidelines.

You will see from the reviewers' comments that while Reviewer #1 has only requested minor revisions, Reviewer #3 is more concerned with the framing of the paper and thinks that its primary focus should be on the role of plasticity in reproductive isolation. I can say that during our initial assessment of your manuscript, this is also the finding that we had found most interesting from an editorial perspective. We do not expect you to restructure the whole paper but we do expect minor changes in various sections (especially in the title and abstract) so that the role of developmental plasticity in reproductive isolation is better emphasized, as we think that this would improve the appeal of your paper to the NEE audience. As an example of the revisions we would like to see in the final paper, here is my suggestion for an alternate title: "Meta-analysis reveals that developmental plasticity promotes reproductive isolation during incipient ecological speciation". If appropriate, 'accelerates' would also work instead of 'promotes'.

Next steps: If the current version of your manuscript is in a PDF format, please email us a copy of the file in an editable format (Microsoft Word or LaTeX)-- we can not proceed with PDFs at this stage. We will perform detailed checks on this file and send you a checklist of our editorial and formatting requirements within about a week.

Meanwhile, you can start revising the manuscript on your end but please do not upload any final materials to the submission system until you receive the formatting checklist from us. Once the checklist arrives, you can incorporate the (usually fairly straightforward) formatting changes in your revised file, and only then upload finalized versions to the submission system.

[redacted]

Reviewer #1 (Remarks to the Author):

I previously reviewed this manuscript and I'm very satisfied with the careful and thoughtful changes done to revise the paper. The authors have done an excellent job dealing with mine and the other reviewers' comments. Exploration of the interaction between common garden and barrier type is a particularly interesting addition to the paper that really strengthens it and I appreciate the extra analysis. I also appreciate the work put into clearing up the presentation of the results and adding clarity around the Bayesian analysis. I think this paper is much stronger and clearer now and will make an outstanding contribution to the field. Congratulations!

I noted a few minor grammatical things as I re-read the paper that the authors may wish to fix:

I55. Add "be" before "the" where the word "dominant" is.

L111. I think you replaced "grand" with "full", so may want to do that here to be consistent.

All the best,
Daniel Noble

Reviewer #1 (Remarks on code availability):

I think the authors have cleaned up their repository and code well.

Reviewer #2 (Remarks to the Author):

The authors have done an admirable job of addressing all of my previous concerns with their paper (which were relatively minor). Overall, this is an excellent paper that will make an important contribution to the fields of speciation and plasticity.

Reviewer #3 (Remarks to the Author):

I'm still not fully convinced by the interpretation. It seems to me that the most parsimonious explanation of the results is that reproductive isolation in these experiments was induced by plasticity, with little contribution from genetic changes. Evidence for a role of genetic changes consists of a non-significant trend in the direction of reproductive isolation for studies that used a common-garden generation prior to the isolation assay (Fig. 3A), and the fact that the interaction is not supported (which could reflect insufficient statistical power). The role of genetic changes is directly contradicted by the finding that isolation failed to increase over generations. On the other hand, there's a strong effect of a common-garden generation, indicating a role for plasticity. Taken together, these results provide clear evidence of plasticity-induced isolation, but weak evidence of genetic evolution of reproductive isolation in the experiments included in this meta-analysis. Despite this, the paper emphasizes evidence of the evolution of isolation, and mentions the role of plasticity only as a secondary finding.

I do think that this is an interesting study, and hope that it will generate discussion.

Reviewers' comments:

Reviewer #1 (Remarks to the Author):

I was excited to read this paper. It tackles a fundamental question in evolutionary biology, how does reproductive isolation evolve in the early stages of speciation? This is a crucial question for understanding patterns of speciation and remains a challenging question to completely nail down given that the speciation process is difficult to capture. The authors of this paper set up a creative and innovative way of testing these ideas by meta-analysing experimental evolution experiments to capture the early stages of reproductive isolation. Unsurprisingly, the analysis is mainly restricted to insects and a couple vertebrate systems which are the only systems that are tractable for such experiments. Nonetheless, this shouldn't detract from the important insights we gain from synthesising such empirical work.

The authors beautifully establish the working theory and clearly connect the effect size and moderators in the meta-analysis to that theory. The analyses are robust. I have reviewed a lot of meta-analysis in my career, and I can tell you that this one ranks highly. The systematic searches are done robustly, careful thought has gone into the effect size calculation and its meaning, sensitivity of results carefully explored, and the conclusions justified. While there is still some room for improvement, I'm convinced the findings will remain robust and make a fundamental contribution to the field with broad implications for understanding the speciation process. I provide more detailed comments below. While my comments seem substantial, I think they are all addressable. I hope these will help further improve what I think is an excellent contribution.

Many thanks for your kind words.

Comments:

1) L28-29. Something reads wrong with "This views..." sentence. It sounds awkward as written. May be cut and reword to: "A long-standing explanation for the evolution of reproductive isolation is the ecological speciation model^{4,7,14}, whereby reproductive isolation evolves as a by-product of natural selection operating in different environments^{4,14}"

Done.

2) L30-33. This is a mouthful. I'd suggest breaking this sentence into two. There are too many ideas here.

This has been rephrased to the following:

"Adaptation in response to divergent selection imposed by different environments can result in greater divergence in inherited factors, like alleles or symbionts, between populations than populations experiencing similar selective regimes. This increased divergence between populations increases the likelihood that other inherited factors accelerate the rate of evolution of reproductive isolation between the populations. For example, barrier loci may hitchhike with alleles at other loci that are subject to divergent natural selection in the different environments^{4,7}. Alternatively, selected loci could pleiotropically promote reproductive isolation^{4,7}." (L33–39)

3) L37-40. Interesting hypothesis. I take it that this is dependent on levels of gene flow and population size? This isn't really mentioned. Small effective population sizes would result in genetic drift dominating which would fix different mutations in each population by chance, that makes sense to me. But wouldn't this also make selection less effective? The statements around this are unclear. It is almost implied that selection is important when stating: "also driven by selection", but then how can this be if you assume selection pressures are the same (as stated on L38)? As it stands, it's a little vague on the exact processes and how this plays out. I think some more detail on this hypothesis

(which I'll admit I am not familiar with) is warranted. I suggest some rewording and adding some text to improve clarity.

The level of gene flow and population size will undoubtedly impact evolutionary dynamics and how fast reproductive isolation evolves between populations. Mutation-order speciation and ecological speciation differ in the environments that populations are experiencing—the same environment, or different environments, respectively. The probability of mutations being fixed is largely a stochastic process, with small population sizes increasing the effect of genetic drift. We have amended this section to the following:

“An alternative mechanism, also driven by selection, is mutation-order speciation, whereby populations experience similar selection pressures, but adaptations may be underpinned by the same, or by different, mutations in the different populations. The order in which mutations fix in each population can then contribute to the evolution of reproductive isolation as these mutations may be incompatible with each other, causing hybrid inviability or sterility via Dobzhansky-Muller incompatibilities^{1,6}.” (L43–48)

4) L40-44. I think some more detail on this study is needed because everything stated here is a little vague. Its meaning is unclear without a bit more context around the study.

The description of Anderson and Weir (2022) has been expanded to the following:

“Species adapting to similar environments are predicted to show similar phenotypes in ecologically relevant traits, like body size and mouthparts; a prediction used by Anderson & Weir¹⁷ in a recent comparative analysis designed to quantify the relative contribution of ecological speciation and mutation-order speciation mechanisms in a number of vertebrate species. They found that sister species were about ten times more likely to be phenotypically similar to each other, which the authors interpret as evidence for sister species evolving in similar environments¹⁷. This intriguing result suggests that mutation-order speciation could be the dominant speciation mechanism in vertebrates and ecological speciation might only explain a minority of speciation events¹⁷.” (L48–56)

5) Fig 1B. Overall, figure 1 is a nice figure but I struggled to understand what was trying to be shown in the panels of Fig 1B. The text was more useful but there needs to be more explanation of the ‘grid’ and what A and B are referring to, why are some boxes blank? I think there is clarity on L577-580, but this should be detailed in the legend as the figure should stand on its own.

We wanted to show something quite simple—that there are two possible crosses: within-environment and between-environment. We can see that the grid is confusing and have replaced this (panel B) with the between-environment crosses. Panel A now shows the within-environment crosses.

6) L96. ‘in’ for ‘n’

Done.

7) L158-159. I think this is the context that is needed in the methods to better set up the reasoning behind analyses. I suggest repeating briefly in the methods. (L710)

Done.

8) L164-166. Probably also worth pointing out that the slope estimate is negative, even if not significant, and with a reduced dataset you lose power, with this subset I think, based on what you state on L158-159, your prediction would actually be a negative slope, right? If so, pointing out that this slope is actually going in the predicted direction is important.

We have mentioned the slope is negative and would fit with the prediction made earlier in the paragraph. It now reads:

“The average slope was not significantly different from zero in this subset of studies either (slope estimate = -0.224 , 95% CIs = $[-1.046, 0.526]$, pMCMC = 0.54), but the slope estimate tended to be negative, which is consistent with reproductive isolation emerging early in the experiments when selection is strong and populations are still relatively maladapted.” (L181–185)

9) L179-180. I find it challenging to understand this point fully. I am not doubting that plasticity can drive phenotypic differences, but I'm surprised at the notion that such effects can drive reproductive isolation in a general sense as implied here. I can understand this for pre-mating isolation mechanisms but not post-mating. Can the authors expand on the references with a couple more sentences outlining the mechanism in a little more detail? Are there empirical examples where post-mating isolation mechanisms are involved? Or do they really mean with respect to pre-mating isolation mechanisms and not post-mating? That would make more sense to me, but then, it should be clarified here. Is there sufficient data to test the hypothesis that this is only for pre-mating isolation mechanisms by adding an interaction between the isolation barrier moderator and whether it was a common garden? That would test this hypothesis directly. It may show me that I'm wrong! I realise, however, that there may not be enough data to test this interaction, but it would be an interesting addition because the intuition that this applies to pre- and post-mating isolation mechanisms may not be clear to readers. On L205-213 the authors argumentation seems to support my point as much of what is discussed seems to be pre-mating.

This is an excellent point. It is an idea that makes intuitive sense for pre-mating isolating barriers to be more influenced by phenotypic plasticity than post-mating barriers. We have included this observation in this section. In addition, we performed the interaction and found a significant interaction: plasticity only increases pre-mating barriers, as you predicted. We have thus altered text throughout and combined Fig 3 and 4 into a new figure to exhibit the interaction.

10) L234. I wouldn't be hesitant to suggest causality. I would suggest simply removing it. It's not needed in any case.

Experiments are the gold-standard for evidence of causality and so we would like to reject this suggestion. The experiments in our meta-analysis are indeed causally testing this hypothesis; that taking a single population, splitting it into replicates, and experimentally imposing divergent selection is testing the causal link between the divergent selection and the accumulation of reproductive isolation that these population pairs exhibit.

11) Fig 2a, 3, 4. Can you plot the line at 0 above the points? The yellow/red dots are covering it making it hard to match with the 95% CIs. Alternatively (and possibly in addition), adding some text with the mean effect and 95% CI would be good as it would provide the exact mean estimates and uncertainty. I'm aware that these are in the text, but I like these on the figures personally because there is no ambiguity.

We have plotted the 0 line above the points and added the mean with 95% CIs as requested.

12) L507-508. Inclusion criteria 2 is awkwardly worded. Do you mean “live” organisms and “any taxa” rather than “taxon unspecific”.

We do, and have edited this sentence to now read:

“the study needed to include data from live organisms from any taxon;” (L525–526).

13) L512-514. How was criteria 6 evaluated practically? I'd be clear on that because it may be that some studies didn't measure and present selection gradients. Would that be true?

Some experiments artificially selected on assortative mating, or on putative sexual traits that are directly linked to reproductive isolation. We have added this information: “selection did not directly act on reproductive isolation itself (e.g. studies were excluded if artificial selection was performed on assortative mating or on a putatively sexual trait).” (L530–532).

14) L516. What were your measures of reproductive isolation? Be specific here. ON that note, this should also be detailed more on L518; For example, “via any isolating barrier”, what do you mean specifically by this?

We have included specific examples of what measures we extracted and expanded on what isolating barriers were present in our data: “Our PICO components were: Population, experimentally-evolved living organisms; Intervention, experimental evolution in different environments and with measures of reproductive isolation between populations (e.g., mate choice including latency to mate or interact, habitat choice, fecundity, longevity after mating, hybrid fitness); Comparison, estimates of reproductive isolation between populations evolving in the same environment compared to populations evolving in different environments; Outcome, estimate of reproductive isolation acting via any isolating barrier (e.g., pre- or post-mating barriers).” (L532–538)

15) L547. Change “out” -> ‘our’

Done.

16) L561-600. This metric makes sense, and I agree with what the authors have done, but I find the workflow odd. How they have written the description of the results creates a lot of conflicting information and leads to uncertainty about what was done and why. It wasn't until reading section 7.5.1 that I realised the authors are actually aware of the utility of the Bayesian approach they are taking, and this section was re-assuring to me because it shows the results were the same. I say this because from my experience in applying the same types of approaches the authors are taking here, sampling variance computed from the Bayesian methods versus those calculated from the sampling variance equations (formula for Z_r) can be quite different. In essence, the transformation from the effect size in each individual study script to Z_r could be done with the whole posterior distribution, and then, using that, the authors can take the SD of the posterior distribution of the effect size (which is the sampling standard error). Then, square it to get the sampling variance. Now you have both the effect size AND the sampling error directly from your Bayesian models. This is what section 7.5.1 is doing but this important point does not come across. I'd trust the Bayesian sampling variances more than the Z_r equation because the models used to calculate effects are binomial, beta etc. What I find odd in how this is written and presented is that, despite being aware of this (as suggested in 7.5.1) the authors opt to calculate the sampling error according to the typical sampling error for Z_r [i.e., use the point estimate from the Bayesian models then use “z_transform” AND $1/(n-3)$ to calculate the effect and its sampling error] as indicated in the first part of script “0. Overview.R”. Why? It's good that these two methods say the same thing, but it does create a lot of confusion and defeats the purpose of using the raw data in the brms analyses. My suggestion would be to present only the results based on the sampling variances taken from the posterior distributions of each effect size in the main MS. Then, run the meta-analysis on that. It's more logically coherent with the Bayesian analyses the authors are using, is more accurate, and it means that the writing can remain focused and clear. I think it's more strongly justified as well. I'm assured given section 7.5.1 that this won't change the overall conclusions, but it will make the paper much stronger in presentation and will be analytically more coherent.

We went back and forth with this when analysing the data and are glad to take the reviewer's advice. We now use sampling variances obtained from our Bayesian models as suggested and have moved the $1/(n-3)$ results to the methods. We originally used the $1/(n-3)$ sampling variance

as it was easily calculable and did not require the raw data to do so. Many of the papers that contributed to this meta-analysis were published before uploading raw data was mandatory. We adopted both approaches to show that the Bayesian methods and point estimates with $1/(n-3)$ yielded the same results. We have made this explicit on L103–106.

17) L597-600. Great way to creatively use the means and SDs to generate analogous distributions to your Bayesian ones! We've used a very similar approach (Noble et al. 2019. PNAS. 116, 13452–13461; Radersma et al. 72. Evolution Letters 4, 360–370) using the multivariate normal. I wonder if the authors could just add a sentence to describe what distribution(s) was used for simulation? Beta-binomial? Poisson?

We have added in the requested information. It now reads:

“In cases where authors did not respond, we extracted means and errors from figures using the R package “metaDigitise”¹¹⁴, randomly generating data with the same parameters using an appropriate distribution (normal, truncated normal, and Poisson were used), and estimated RI from these (83 / 1723 effect sizes), a method used by Noble et al.¹¹⁵ Effect sizes calculated this way are coded as “1” in the “backtransformed” column of the data.” (L618–622)

18) L672-675. Great to be thorough on the publication bias but many of the tools used are not appropriate, as pointed out by the authors (L674). My suggestion would be to stay focused on the regression approaches that include as moderators time lag and sampling variance / effective sample size. Report and correct effects on these. This section and details on 660-663 can simply be removed or just moved to supplement. Saves you space. It's important that the slope is presented from your multi-level meta-regression, but I'd also suggest adding the corrected effect size because the intercept from such a model corrects the overall effect size (technically) for publication bias.

We included the funnel plot, trim and fill, and fail safe number analyses for completeness as they might prove useful to future researchers evaluating e.g. the quality of meta-analyses or bias in these metrics. We now report the intercept from our multi-level meta-regression model evaluating publication bias, which provides an estimate of the adjusted mean effect size (Nakagawa et al. 2022).

19) L681-682. Remove the heterogeneity from the random effects model. It's not the appropriate model for these data.

We are not sure we follow. The ‘random’ effect model here only includes the between-study heterogeneity (Tau). We then contrast this with a model which includes other sources of heterogeneity (repeated measures on different species, research group ID and phylogeny). This is for the benefit of other meta-analysts who may be interested in how variance is partitioned among these different sources of heterogeneity, following recommendations in Nakagawa & Santos (2012).

20) L689. Did you run only a single chain? I'm fine with this if the chains are mixing well and they are run long enough, but I'd suggest adding some justification here as many folks using Bayesian estimation procedures insist on multiple chains. Multiple chains can be useful and important to ensure that chains are converging to same answer as the MCMC process is inherently stochastic in its starting values and the ways in which it explores parameter space. I'd also like to be re-assured that chains are converging because when using the posterior distribution for the sampling variance it could matter.

We performed three chains with different initial conditions for each model. Visual inspection and Gelman's diagnostic tests confirmed all chains converged. We have added information concerning the number of chains run for each Bayesian model.

21) L692. “full model” instead of “grand model” is more commonly applied terminology?

We have changed this.

22) L687-708. I'd suggest making changes to the presentation and reporting of mean effects from this model to make them more easily interpretable and meaningful. This would also require some updates to the figures (sorry), but I think it will be worth it. It doesn't appear that you centred generation time on a biologically sensible number of generations, say, 10? If not (appears not) I'd suggest doing that because it will make your intercepts more meaningful. Alternatively, you can also calculate conditional mean estimates for different generation times for each of your categorical moderators (e.g., common garden, isolating mechanism etc) which would make your mean estimates in Fig 3 and 4 also easier to interpret. We discuss some thoughts on this in Noble et al. 2022. *J Experimental Biology*. 225, jeb243225 in a slightly different context but one that is nearly equivalent. The idea is that, if you centre generation time on something meaningful then you can be more precise about what the mean effect sizes are in your figures (2, 3, 4). In other words, they would be the mean effect size after, say, 10 generations. In fact, I would recommend these figures present multiple means at different generation times (i.e., condition the means on generation time) to demonstrate how means change through time. It would also imply that you can simply remove Fig 2 B because it would be wrapped up in the means at different generation times anyway. To be clear, I suggest having a look at this paper O'Dea et al. 2019. *Fish and Fisheries*, 20: 1005-1022 (fig 6.) and also we show how to use orchard package to do that here: <https://danielInoble.github.io/orchaRd/#example-6-meta-regression-and-conditional-mean-effect-sizes>.

We appreciate these suggestions and used the suggested code to produce the recommended visualisations. However, given that generation time did not influence the evolution of reproductive isolation, all means were equivalent, making the plots appear too busy without the addition of greater interpretation. Further, Plot 2B, which shows the relationship with the number of generations, is a good tool to display the distribution of the data that comprises our dataset.

23) L710 onwards: Sorry, I'm struggling to understand what the authors did here. It's unclear why subsetting the data to only studies estimating RI more than twice is useful from these few introductory sentences. I interpret it being useful because the authors have some studies that provide an effect size at more than one time point across generations. For example, for a study that does, say 30 generations, they measure RI, after say, 2 generations then test RI (giving you one effect size), then let them evolve for another 2 generations and then again measure RI. Giving you a second effect size. Is this the right way to think about it? If so, I'd suggest writing some of the logic / justification to explain this better. Maybe a clearer topic sentence stating the objective more clearly will help: "To understand XX we did YY". I think some of these points are better discussed in the main body of text, but it should come across down here too. I'm also unclear on what you mean by 'two-stage' model. It seems to me you fit a model on this subset with the structure: effect size ~ generation time. Is that true? If so, then doesn't the intercept just tell you what RI is at generation time 0 (i.e., just the start)? I'm sure this is not what the authors mean but some clarity about the model structure here is needed. Also, why the difference between MCMCglmm and metafor? Did you fit a random slope model in metafor? It's not the same as what MCMCglmm is doing so that could explain the difference. Either way, I think some more work on this section is needed to clarify the goals, predictions and the model structure. That will make the significance of the sections result become clearer to the reader.

The reviewer's interpretation is spot on and we have reworded this section to be more explicit about what we did (L738–768). We used two approaches because they each have strengths and weaknesses. The 'two-stage' approach first fits a linear model to each species separately (to get the intercept and slope) and then combines these in a meta-analytic model (each parameter is weighted by its inverse sampling variance) to get the mean intercept and slope across species. This is an intuitive model which we implemented in metafor (see https://www.metafor-project.org/doku.php/tips:two_stage_analysis). The random slopes model also estimates the mean intercept and slope across species but in a mixed model framework and is data hungry. We fit this model in MCMCglmm as it is not currently possible / undocumented in metafor. The

two approaches tell the same story and give similar parameter estimates (although the CI around the random slope intercept is wider).

24) L717: “The” rather than “Our”

Done.

25) L760-762. Again, I’d suggest the authors give some rationale about what the problem is to justify these analytical choices. From the simple topic sentences here, it’s not clear why they even need to conduct these additional analyses. What’s the problem they are trying to circumvent? Their main analyses are robust, and they seem to be concerned about running additional analyses that (I would argue) are less correct and thus likely to show conflicting results. It’s unclear what they are trying to achieve in such analyses so at the very least some stronger justification is required.

We have removed this analysis as requested. We aimed for completeness, especially considering that data from before 2010 was usually not provided in raw data form. Such an analysis would therefore bolster our results to claims that different ways of analysing the data would lead to different conclusions.

26) Table S5. I can see from this table the authors are using default metafor options. In this case I don’t think it will matter, but I would personally suggest using test = “t” so that the default test statistics is using the t-statistic and not the z-statistic. In addition, considering some simulations we have run (Nakagawa et al. 2022. Ecology 103, e03490) I’d also suggest using the dfs = “contain” argument. Wolfgang implemented this after we did these simulations and will correct the degrees of freedom to be more sensible when you have a lot of within study effects. This should also converge more to line up with your Bayesian models.

We have implemented these changes, thank you for pointing them out.

Hopefully my comments are useful. If anything is unclear do not hesitate to get in touch.

Daniel Noble

Reviewer #1 (Remarks on code availability):

27) Yes, the code has a README detailing the files and the data. Repository could be better organised so that all R script associated with each paper to extract the raw data is found within the PaperData/ folder and inside each papers folder. Otherwise, it’s a little chaotic with so many files. Not totally bad because it can be reproduced still but it is a little overwhelming. I suggest the authors use `pacman::p_load()` function for installation of packages as code will fail if authors have not installed libraries. They should have a single call to `install.packages("pacman"); library(pacman)`, then they can list the packages.

We have moved the paper-specific R files to their respective folders and used pacman for library installing.

Reviewer #2 (Remarks to the Author):

This manuscript presents the results of a meta-analysis in which the authors examined the role of divergent natural selection and phenotypic plasticity on the evolution of reproductive isolation. The authors’ principal findings are: 1) the evolution of reproductive isolation by divergent selection is common and detectable across a range of taxa and environments, and 2) reproductive isolation does not appear to increase with time in experimental speciation studies. The first finding is consistent with ecological speciation theory. The second finding was unexpected, as it has long been assumed that

reproductive isolation should increase with how long two populations have separated. The authors evaluate four alternative hypotheses to explain this surprising result, concluding that it reflects the action of phenotypic plasticity. Specifically, they suggest that divergent environments trigger a plastic increase in reproductive isolation in the first few generations.

Overall, I enjoyed reading this paper and have no major concerns. The data from the meta-analysis alone makes this paper worthwhile. I also appreciated the authors providing suggestions for future research at the end of their paper.

However, I have a few minor suggestions for the authors to consider. They are listed below in the order in which they appeared in the paper.

28) 1) The first line of the Abstract should probably be rewritten. I suspect that many speciation researchers might disagree with the authors' claim that "the factors that shape its [reproductive isolation's] evolution during the early stages of speciation remain largely unknown." How about saying something like "the factors ... speciation require further clarification"?

We have edited the sentence to now read:

"The evolution of reproductive isolation is a key evolutionary process¹⁻³, but the factors that shape its development in the early stages of speciation require clarification." (L12-13)

29) 2) Given the nature of this paper, it would be helpful to provide a general definition of reproductive isolation right up front. If there are slightly different definitions, the authors could provide a table listing some commonly used definitions.

We have included a more direct definition of reproductive isolation:

"Rooted in the Biological Species Concept^{2,3}, reproductive isolation can be viewed as an emergent property of an interaction between two populations, that results in a restriction of gene flow between the same populations²." (L26-28)

30) 3) On page 1, line 31, the authors introduce the idea that adaptation in response to divergent selection sets the stage for the evolution of reproductive isolation by resulting in genetic divergence between populations experiencing contrasting selection pressures. This claim is well supported theoretically and empirically, of course. Still, I wonder if this statement could be made more general by noting that divergent selection can result in divergence between populations IN FACTORS THAT CAN BE INHERITED. This would make the scenario outlined here more general, including cases in which reproductive isolation is mediated via learning or the transmission of symbionts.

We have changed the language such that differences between populations need not be genetic, but also symbionts and other factors that can be inherited. For example, on L33-35, and L136-137.

31) 4) On page 2, line 40, I recommend providing a quick synopsis of mutation-order speciation.

We have edited this section to the following:

"An alternative mechanism, also driven by selection, is mutation-order speciation, whereby populations experience similar selection pressures, but adaptations may be underpinned by the same, or by different, mutations in the different populations. The order in which mutations fix in each population can then contribute to the evolution of reproductive isolation as these mutations may be incompatible with each other, causing hybrid inviability or sterility via Dobzhansky-Muller incompatibilities^{1,6}." (L43-48)

32) 5) Page 3, line 96: typo: "n" should be "in" (as IN Rice and Hostert)

Done.

1) 6) Page 4, lines 111-112, I could not find any explanation in the Methods regarding how the authors defined “same” versus “different” environments. Was this left up to the researchers of the original studies? Was there perhaps an effect on the evolution of reproductive isolation in how different the two environments were to each other that the authors of this study could analyze? This seems to be an important question that a meta-analysis like this could resolve.

In all the papers, the experimenters including replicate populations that were evolving in the same environment, and it was easy to determine which crosses involved populations of the same environment and those of different environments. We would have loved to quantify the differences between environments and somehow use that metric to include as a covariate. It is, however, very difficult to understand if a 5C difference between a cool environment and a hot environment is equal or more or less different than a change in diet, for example. If all the environments were of temperatures, or different host plants, then we can imagine such a metric being easy to quantify and interesting to include in an analysis such as ours. Indeed, some environments differed in many environmental dimensions, and some in few. Further work on this aspect on the evolution of reproductive isolation is therefore much needed.

33) 7) Page 4, line 122: “genetic” => “inherited” (see point #3 above)

Done.

34) 8) Page 5, line 157: The authors note that the experiments they reviewed for their meta-analysis imposed “a large magnitude of selection” on the experimental subjects. Can this magnitude of selection be quantified (or a range of values reported) to justify this claim?

In an ideal world, we would very much like to tie in the magnitude of selection on the evolution of reproductive isolation, but such data are lacking for the majority of studies. We have revised this statement accordingly:

“One possible explanation for this unexpected result is that the environmental manipulations designed by experimenters induced strong selection in the early stages of the experiment so that adaptation and concomitant reproductive isolation developed rapidly and almost instantaneously.” (L171–173)

35) 9) Page 5, line 179: I would use the term “phenotypic plasticity” (rather than “developmental plasticity”) throughout the paper since the former is a more widely used and general term. This paper does not need to distinguish between these two terms.

Done.

Overall, this is a nice manuscript that I hope to see in print soon!

Thank you!

Reviewer #3 (Remarks to the Author):

This study uses a meta-analytic approach to investigate the role of divergent selection in distinct environments in the evolution of reproductive isolation. This is a very interesting question. It’s widely assumed that reproductive isolation evolves passively through accumulation of mutations that induce incompatibility via pleiotropic effects on reproductive traits. However, ecological speciation theory suggests that divergent selection can accelerate this process. The relative importance of these processes is not clear. This study addresses this question directly, and provides some novel insights. Nonetheless, I have a few suggestions and would like to see some additional analysis to clarify the

findings.

36) To me, the most interesting finding was that plasticity seems to contribute a great deal to reproductive isolation. Indeed, the “difference estimate” for the role of plasticity (~ -0.1) was considerably greater than the “difference estimate” for the comparison of populations evolving in different vs. same environments (~ -0.07). This left me wondering whether there’s any evidence that divergent selection contributed to isolation in these experiments, once the effect of plasticity is factored in. The data shown in Fig. 3 suggest that there is not: the mean effect size for studies that incorporated a common-garden step prior to isolation assays does not appear to be different from zero. It’s thus potentially misleading to conclude that divergent selection promotes isolation based on an analysis that ignores the role of plasticity. To clarify this, it would be helpful to report separate analyses for studies that subjected their experimental animals to a common-garden step prior to isolation assays vs. studies that did not do so. Better yet, perhaps a model incorporating both environment (same vs. different) and plasticity (presence vs. absence of a common-garden step) could be used and the interaction of environment x plasticity could be tested. If this analysis shows that there’s no evidence that divergent selection promotes isolation once the role of plasticity is factored in, that would be a very interesting finding!

We apologise for our lack of clarity on this matter. The results we reported throughout the manuscript were from a full model (without interactions) that included both the effects of a common garden environment and the effects of the different vs. same environment. The effects did not mask one another. For completeness, we also included a model of the interaction between the common garden factor and the same vs. different treatment. There is no evidence for an interaction between the two, however. We include this result on L724.

Other comments:

37) L17: Provide a median and range for the number of generations here.

This information is provided in the main text (L116).

38) L28: This is an odd word to use here. Perhaps change to “posits” or “proposes”?

This has been edited to the following:

“A long-standing explanation for the evolution of reproductive isolation is the ecological speciation model^{4,7,14}, whereby reproductive isolation evolves as a by-product of natural selection promoting different phenotypes in different environments^{4,14}.” (L30–33)

39) L53: What is a “speciation continuum”?

One thing we were acutely aware of when writing this paper was how speciation can be studied at the microevolutionary level and from the macroevolutionary perspective as we, the authors, cross these two ends. Many others have also written about reconciling both micro- and macroevolutionary perspectives on speciation and the speciation continuum is one way to conceptualise this. Throughout, we have referred to the speciation process instead.

40) L96: Typo (“as n Rice and Hostert”)

Done.

41) For each test, it would be helpful to state the number of effect sizes in the text.

Done.

42) L202-203: I wasn’t sure what point the authors were aiming to make here. If you're thinking about

populations inhabiting different environments that impose differential selection, then surely there will be consistent differences between those environments over many generations. Are you dismissing the possibility of sympatric speciation initiated by development in distinct habitat patches?

We are not dismissing it, merely pointing out that if offspring produced by each environment randomly disperse to either environment, then the build-up of reproductive isolation stalls.

43) Figure 1: Add sample sizes for each taxon to panel D.

Will do.

44) Looking at distributions of effect sizes in Figures 2, 3 and 4, there appear to be large differences in both the number of effect sizes and their variance between the groups being compared. For example, the number of effect sizes for within vs. between-environment assays is very different, as is the variance of effect sizes. Does this require a correction for unequal variances, or is such a correction already incorporated into the analysis?

Bayesian multi-level models are relatively robust to unequal variances and differences in sample size. Fitting models to test for unequal variances across multiple covariates is quite data hungry, and so we compared univariate models that had a fixed variance with a model that allowed the variance to differ. For all our covariates, we found this did not change our interpretation of the model. The code for this has been included in the R script 7.3_main_mcmcglmm.R.